# Exercise as Osteoarthritis Treatment in Wistar Rats Promotes Frequency-Dependent Benefits

**DOI:** 10.3390/biology14111537

**Published:** 2025-11-03

**Authors:** Mateus Cardoso Colares, Anand Thirupathi, Leandro Almeida da Silva, Daniela Pacheco dos Santos Haupenthal, Laura de Roch Casagrande, Ligia Milanez Venturini, Yaodong Gu, Camila da Costa, Igor Ramos Lima, Vitória Oliveira Silva da Silva, Luciano Acordi da Silva, André Domingos Lass, Ricardo Aurino Pinho, Paulo Cesar Lock Silveira

**Affiliations:** 1Faculty of Sports Science, Ningbo University, Ningbo 315211, China; mateuscc@gmail.com (M.C.C.); guyaodong@nbu.edu.cn (Y.G.); 2Laboratory of Experimental Physiopathology, Program of Postgraduate in Science of Health, Universidade do Extremo Sul Catarinense, Criciúma 88806-000, Santa Catarina State, Brazil; leandro.silva@ifsc.edu.br (L.A.d.S.); danielapshaupenthal@yahoo.com (D.P.d.S.H.); lauracasag@outlook.com (L.d.R.C.); ligia_milanez@hotmail.com (L.M.V.); camiladacosta49@gmail.com (C.d.C.); igor.mamp@unesc.net (I.R.L.); silvadasilva.vi@gmail.com (V.O.S.d.S.); luciano_acordi@unesc.net (L.A.d.S.); 3Graduate Program in Health Sciences, School of Medicine and Life Sciences, Pontifícia Universidade Católica do Paraná, Curitiba 80215-901, Paraná State, Brazil; andrelass0974@gmail.com (A.D.L.); rapinho12@gmail.com (R.A.P.)

**Keywords:** osteoarthritis, exercise, treadmill, inflammation, oxidative stress, cartilage

## Abstract

Osteoarthritis is a disease that causes joint pain and difficulty moving, affecting millions of people worldwide. This study investigated how different frequencies of moderate treadmill exercise could aid in the treatment of osteoarthritis in rats. The treatment consisted of exercising three or five times per week. The results showed that both exercise regimens helped inhibit the progression of the disease, but exercising five times per week provided more significant benefits, such as higher energy expenditure, weight reduction, decreased inflammation, and better cartilage protection. This suggests that exercise frequency is important for optimizing outcomes. These findings indicate that regular exercise, especially at higher frequencies, may be an effective and accessible strategy for managing osteoarthritis, improving patients’ quality of life.

## 1. Introduction

Osteoarthritis (OA) is a chronic, multifactorial joint disease characterized by cartilage degeneration, inflammation, oxidative stress, and bone changes [1,2,3,4]. Physical exercise is one of the main therapies recommended for the treatment of OA [5,6,7,8,9]. It can act on several factors such as weight control, muscle strengthening, and modulation of inflammation and redox state [6,10,11].

However, it is necessary to consider the characteristics of an exercise program, such as type, intensity, duration, and frequency of sessions. As demonstrated by Iijima et al. [12], animals with induced OA showed a lower degree of injury when performing moderate exercise compared to untrained animals or those that performed intense exercises. Yamaguchi et al. [13] reported in an experimental study that moderate exercise suppressed the progression of OA, while intense exercise increased the damage. Excessive exercise can increase mechanical and metabolic stress in the joints, favoring the onset of microinjuries, inflammation, and oxidative stress, thus aggravating the clinical picture of OA [12,14,15]. However, moderate-intensity exercise improves blood circulation and cartilage nutrition, as well as promotes anti-inflammatory and redox adaptations [6,11,16,17,18].

In addition, within the discussion of the beneficial effects of exercise, the maintenance/chronification of physical exercise is listed as an important point for obtaining favorable effects [19,20,21,22,23,24]. That said, the frequency of exercise execution can be determinant to potentiate protective effects in the treatment of OA [25]. Thus, in mid-2021, a search of the PubMED-MEDLINE database was conducted using MESH descriptors such as “Osteoarthritis AND Exercise” and filters such as “Other animals” and “10 years”, yielding 81 articles. After reading the titles and abstracts, 43 articles were selected. After reading the full text, 38 studies met the eligibility criteria (articles that associated animal models of knee OA with physical exercise over the last decade). It was found that none of the 38 studies found analyzed the effect of different weekly exercise frequencies on OA [11,21,22,23,24,25,26,27,28,29,30,31,32,33,34,35,36,37,38,39,40,41,42,43,44,45].

The same lack of information on frequency was found in review studies on the same topic that evaluated articles from before the last decade. In these reviews, the main outcome was based on exercise intensity [46,47,48,49].

That said, previous studies have focused more on intensity, bringing to light valuable information [46,47,48,49], but leaving incomplete the understanding of the effects of exercise since the frequency of sessions can also be a relevant factor. Thus, this study aimed to evaluate the effects of exercise frequency on the treatment of experimental OA. In addition, it is hypothesized that moderate exercise may favor a protective effect on the joint, especially at a frequency with a greater number of sessions given the greater number of weekly activity stimuli.

## 2. Materials and Methods

### 2.1. Animals

Initially, 60 male Wistar rats were used, divided into 4 to 5 animals per cage; the rats were 2 months old and weighed between 200 and 300 g. These boxes (cages) were approximately 30 cm long × 20 cm wide × 13 cm high. They had a wood shaving lining the interior base and a metal grate on top for air circulation, inspection, and storage of food and water bottles.

The cages were kept in a controlled climate of 21 ± 1 °C, with a 12/12-h light/dark cycle. Rats were provided standard rodent chow and water from the institution’s hydraulic system ad libitum. The study was permitted by the Animal Ethics Committee of the main institution under approval 083/2021, conducted in accordance with Brazilian guidelines for use of animals for scientific and didactic purposes, and based on compliance with the ARRIVE 2.0 guidelines [50].

Based on the instructions issued by animal ethics committees, animal research recommends the use of the 3Rs: Replacement, Reduction, and Refinement [51]. Thus, to use the minimum number of animals necessary to obtain the results, the number of animals was based on a review of studies with animal models and exercise [4,6,10,11,12,16,19,21,22,23,24,26,27,28,29,30,31,32,33,34,35,37,38,39,40,41,52,53,54,55,56,57,58,59], for the possibility of a difference of up to 17 to 25% in the parameters to be analyzed between the groups, with a variance of up to 10%, calculated with the EDA tool [52,60]. This results in a sample size of 14 animals per group (6 to 9 animals for metabolic activity analyses, ELISAs, and biochemical tests and 5 animals/group for histological analyses).

The rats were numbered and randomly divided, by drawing lots, into 4 groups containing 14 to 16 animals: control (sham): *n* = 14; osteoarthritis (OA): *n* = 14; osteoarthritis + moderate exercise 3 times per week on alternate days (OA + 3×): *n* = 16; and osteoarthritis + moderate exercise 5 times per week on consecutive days (OA + 5×): *n* = 16. The intervention groups were increased by 2 animals per group ((14 + 2 = 16) = (+14%)) to compensate for possible losses or non-adaptation to the exercise protocol.

Not all animals adapt to walking on the treadmill, resulting in their exclusion from the experiment. If there were no exclusionary circumstances for the additional animals, such as suffering and/or lack of adaptation to the exercise, they would be kept in the study counts. However, of the 60 animals involved in the experiment, 2 in each exercise group were excluded due to inability to perform treadmill walking movements. Thus, the results presented are organized from a sample size of 56 rats subdivided equally into 14 animals per group. The excluded animals were not used in the analyses.

### 2.2. Osteoarthritis Model

The model proposed by Yamada et al. [53] was used to induce OA. The procedure was performed with a single intra-articular injection of 1.5 mg of sodium monoiodoacetate (MIA) (Sigma-Aldrich corporation, lot: #SLCK0393, Saint Louis, MO, USA) diluted in 50 μL of 0.9% saline solution [53],at a concentration of 30 mg/mL.

The animals were anesthetized with 4% isoflurane via inhalation until the corneal-palpebral reflex was absent and movement was absent. After being anesthetized, they were placed in the supine position, their knees were flexed and trichotomized, aseptically cleaned, and the MIA compound was administered through the medial region adjacent to the infrapatellar ligament of the right knee (Figure 1). The sham group underwent the same procedure but using only 0.9% saline solution. The applicator was blinded throughout the model induction.

The MIA preparation was performed on the day of OA induction and stored in a falcon tube for the subsequent application procedure. Syringes with 4 mm fixed needles and a capacity of 500 μL, commonly used for insulin administration, were used. To better control the amount injected, 50 μL of the preparation was aspirated from the Falcon tube into the syringes, and the syringes were changed after each injection.

### 2.3. Exercise Protocols

For the exercise sessions, a motorized treadmill insight (Insight, model EP-131, Ribeirão Preto, SP, Brazil) was used, consisting of 6 individual stalls with a height of 15 cm, an internal width of 10 cm and a length of 50 cm (Figure 2a). The equipment had electronic speed, time, and electrostimulation regulators, all controlled by a digital panel that displayed speed (m/min), time (minutes), distance (meters), and amperage (mA). The tilt was manually adjusted and remained steady throughout the experiment. Before the protocols began, the equipment underwent inspection and calibration. No electrical stimulation was used in the exercise sessions; only table tennis balls were used to stimulate the animals by touch and sound to follow the course of the treadmill (Figure 2b).

The exercise protocol, summarized and shown in Table 1, was based on Cifuentes et al. [55], and adapted from analyses of reviews on exercise and animal models of OA [46,47,48,49], in addition to the analysis of the set of information from articles related to exercise and animal models of OA [4,10,11,12,13,14,19,21,22,23,24,26,27,28,29,30,31,32,33,34,35,37,38,39,40,41,42,43,44,45,55,56,57,58,59]. Thus, the protocol for this study was organized in order to be aligned with the main characteristics found in the literature for the use of exercise as a therapy for OA.

The aerobic exercise sessions were conducted from the 15th day after intra-articular injection of MIA (Figure 3), in which all animals in the exercise groups were exposed to 4 days of treadmill adaptation at a speed of 10 m/min, with no incline and with time progression (1st day: 10 min, 2nd day: 15 min, 3rd day: 20 min, and 4th day: 25 min). After the adaptation period, the animals were subjected to training for 8 weeks, with a weekly frequency of 3 or 5 times (Figure 3), performed on a treadmill without incline, 30 min/day, at a speed of 13 m/min for the first 4 weeks of training and a speed of 16 m/min in the subsequent weeks, as adapted from [29]. It should be noted that to eliminate any eventualities that could lead to the interruption of the exercise, regardless of the cause, the animals were monitored by the researchers during all sessions.

### 2.4. Euthanasia and Tissue Preparation

After the physical training intervention, the animals were anesthetized with inhalation of 4% isoflurane and euthanized by decapitation. Subsequently, with the researchers blinded, the intra-articular tissues and the gastrocnemius muscle of the right hindlimb were removed, numbered and stored in a −80 °C freezer for subsequent analysis.

From each group, 6 samples of gastrocnemius muscle were designated for metabolic analysis. For gene expression and molecular evaluations, 6 to 9 samples per group were used from the intra-articular tissues (all intracapsular tissues of the joint). Before the homogenization of these samples, 3 samples from each group were randomly selected for macroimaging of the tibial plateau and femoral condyle and then returned for homogenization. For histological evaluations, 5 samples from each group were used containing the distal femoral bone epiphysis with the cartilage surface, the proximal tibial bone epiphysis with the cartilage surface, and the meniscus.

### 2.5. Analysis

To ensure reproducibility and accuracy of results, all analyses were performed blindly and in duplicate. The values presented correspond to the average of the replicates.

#### 2.5.1. Metabolic Activity

For analysis of metabolic activity, the gastrocnemius muscle from the same leg as OA induction was used (6 samples/group). SETH buffer (250 mM sucrose, 2 mM EDTA, 10 mM Tris base, 50 IU/mL heparin) pH 7.4 was used for homogenization. The homogenate was centrifuged at 3000 RPM for 10 min, and the supernatant was stored in a freezer at −80 °C until analysis.

##### Citrate Synthase

The homogenate was incubated in a medium containing 0.1 mM 5,5′-dithiobis-(2-nitrobenzoic acid) (DTNB), 0.2 mM oxaloacetic acid, 0.1% Triton X-100 (Sigma-Aldrich corporation, Saint Louis, MO, USA) and 0.1 mM acetyl-CoA, in a 100 mM Tris-HCl buffer, pH 8.0. The 2-nitro-5-thiobenzoic acid (TNB), reduced from DTNB was measured spectrophotometrically at 412 nm for 3 min at 37 °C [61]. The activity was expressed as nmol. min^−1^. mg protein^−1^.

##### Succinate Dehydrogenase

To evaluate the activity of the succinate dehydrogenase enzyme, a sample containing 80 to 140 mg of protein was used and the incubation medium containing 62.5 mM potassium phosphate buffer pH 7.4, 0.1% Triton X-100, 1 mM sodium succinate and 9 mM 2,6-dichloroindophenol (DCIP) was added. The systems were pre-incubated at 30 °C for 30 min in a water bath and then 4.3 mM sodium azide, 7 mM rotenone, 1 mM phenazine methosulfate and 42 mM DCIP were added. The reduction in DCIP was determined at 600 nm for 5 min at 25 °C [62]. The activity was expressed as nmol. min^−1^.mg of protein^−1^.

##### Complex I

To the homogenized sample containing potassium phosphate buffer (100 mM, pH 7.4), 14 mM NADH, 1.0 mM rotenone and ferrocyanide (10 mM FeCN) were added. The analyses were performed for a period of 3 min at 420 nm at 25 °C. The activity of complex I was measured by the rate of NADH-dependent ferrocyanide reduction, which can be observed by the decrease in absorbance [63]. The data were presented as nmol/min^−1^.mg protein^−1^.

##### Complex II

The homogenized sample was incubated at 30 °C for 20 min with an incubation medium consisting of potassium phosphate (40 mM, pH 7.4), sodium succinate (16 mM), and DCIP (8 µM). Then, 4 mM sodium azide and 7 mM rotenone were added to the medium, and the reaction was initiated by the addition of 40 mM DCIP. Absorbances were recorded at 600 nm for 5 min. Complex II activity was assessed by the decrease in absorbance caused by the reduction of 2,6-dichloroindophenol [35]. Data were presented as nmol/min^−1^.mg protein^−1^.

#### 2.5.2. Weight Gain Assessment

To weigh the animals, a scale with a milligram scale and a conical containment vessel were used. The container was placed on the scale and zeroed, and then the animals were placed individually in the container for weight assessment. As a follow-up, weighing was performed every week from the beginning of the training period and to assess weight gain, the subtraction of the final weight from the initial weight of each animal was computed. The results were expressed in grams.

#### 2.5.3. Oxidants

Oxidized 2,7-dichlorofluorescein (DCF) levels (n-9 animals/group) were quantified in homogenate samples incubated with 10 mM 2′,7′-dichlorodihydrofluorescein diacetate (DCFH-DA) at 37 °C for 30 min. Quantification of oxidized fluorescent derivatives was performed using a spectrophotometer at excitation and emission wavelengths of 488 and 525 nm. A DCF standard curve was measured using 10 mM DCF as an internal control in the experiment, and the results are expressed as fluorescence intensity (detection limit = 0.01 U fluorescence/mg protein) [64].

NO production (n-9 animals/group) was measured spectrophotometrically through the stable metabolite nitrite. To measure the amount of nitrite, Griess reagent (1% sulfanilamide in 0.1 mol/L HCl and 0.1% n-(1-naphthyl)-ethylenediamine hydrochloride) was used and incubated together with the samples at room temperature for 10 min. The absorbance was read at 540 nm using a microplate reader. The nitrite content was calculated based on a standard curve from 0 to 100 mM performed with the metabolite sodium nitrite. The results were calculated in µmol nitrite/mg protein (detection limit = 0.1 uM/mg protein) [65].

#### 2.5.4. Markers of Oxidative Damage

Total thiol content (n-9 animals/group) was determined using the DTNB method (Ellman’s reagent). Briefly, 30 µL of the sample was mixed with 1 mL of phosphate-buffered saline/1 mM EDTA (pH 7.5). The reaction was initiated by the addition of 30 µL of a 10 mM DTNB stock solution in phosphate-buffered saline. Control samples, which did not include DTNB or protein, were analyzed simultaneously and used as blanks in the final calculation. After 30 min of incubation at room temperature, absorbance was measured at 412 nm, and the amount of TNB (equivalent to the amount of sulfhydryl groups) was calculated (detection limit = 0.001 nmol TNB/mg protein) [64].

Protein oxidative damage (n-9 animals/group) was also determined by quantifying protein carbonyls through the reaction of carbonyl groups with 2,4-dinitrophenylhydrazine (DNPT). 20% trichloroacetic acid (TCA) was added to the samples to precipitate the proteins and incubated with 2,4-dinitrophenylhydrazine. The samples were then redissolved in 6 M guanidine hydrochloride, and the carbonyl contents were determined by reading the absorbance at 370 nm in a spectrophotometer. A molar absorption coefficient of 22,000 M^1^ M^−1^.cm^−1^ was used (detection limit = 0.005 nmol/mg of protein) [66].

#### 2.5.5. Antioxidant Enzyme Activities

SOD activity (n-9 animals/group) was determined by the adrenaline oxidation inhibition reaction. Joint samples were homogenized in glycine buffer. Volumes of 5, 10, and 15 μL of samples were separated after homogenization, and 5 mL of catalase (0.0024 mg/mL in distilled water), 175–185 mL of glycine buffer (0.75 g in 200 mL of distilled water at 32 °C, pH 10.2), and 5 μL of adrenaline (60 mM in distilled water plus 15 mL/mL of fuming HCl) were added, incubated for 180 s, and measured at 10-s intervals on a SpectraMax i3xELISA (Molecular Devices, San Jose, CA, USA) reader at a wavelength of 480 nm. Values were expressed as SOD unit/mg protein (U/mg protein) (detection limit = 0.1 U/mg protein) [67].

Reduced glutathione (GSH) levels (*n* = 9 animals/group) were determined as described by Hissin et al. [68], with some adaptations. GSH was quantified in tissue samples after protein precipitation with 20% trichloroacetic acid (TCA). 800 mM phosphate buffer (pH 7.4) containing 500 mM o-phthaldialdehyde was added to part of the sample. The color development resulting from the reaction between o-phthaldialdehyde and thiols reached a maximum within 5 min and remained stable for more than 30 min. Quantification was performed by fluorescence reading, with excitation at 350 nm and emission at 420 nm after 10 min. A reduced glutathione standard curve was used to calculate GSH levels in the samples. Results are expressed as fluorescence units/mg protein (detection limit = 0.025 U fluorescence/mg protein) [69].

#### 2.5.6. Protein Content Determination

The protein content of homogenized tissue samples (*n* = 9 animals/group) was determined using phosphomolybdic-phosphotungstic reagent (Folin phenol) added to the sample to bind to proteins. The reagent was slowly reduced, changing color from yellow to blue. Bovine serum albumin was used as a standard, according to Lowry [70]. Absorbance was read at 750 nm.

#### 2.5.7. Pro-Inflammatory and Anti-Inflammatory Mediators

For the determination of cytokines (TNF-α, IL1-β, IL6, IL4, IL10, TGF-β), the sandwich enzyme-linked immunosorbent assay (ELISA) method was used, associated with standard enzymes and absorbance readings measured by a spectrophotometer (Invitrogen ELISA kit; Thermo Fischer Scientific—Bender MedSystems GmbH; Vienna, Austria) (detection limit: IL1-β, IL6, and TGF-β = 8 pg/mg protein; TNF-α = 15.6 pg/mg protein; IL10 = 32 pg/mg protein; IL4 = 4 pg/mg protein). Initially, the plate was sensitized for subsequent incubation with the capture antibody. Subsequently, all samples were processed (*n* = 9 animals/group) and incubated overnight. The following day, the detection antibodies, HRP, and substrate were added. At the end of the process, the stop solution is added and the absorbance of the wells is read spectrophotometrically at 450 nm.

#### 2.5.8. Gene Expression

The gene expression (*n* = 6 animals/group) of bone morphogenetic protein (BMP2—Gene ID: 29373) and metalloproteinase 13 (MMP13—Gene ID: 171052) was evaluated. Total RNA was extracted using TRIzol^®^ reagent (Life Technologies, Carlsbad, CA, USA) and following the manufacturer’s recommended instructions. The obtained RNA was solubilized in 30 μL of Milli-Q water, treated with 0.1% DEPC (Sigma-Aldrich corporation, Saint Louis, MO, USA), grouped in a single tube and stored at −20 °C. The total extracted RNA was quantified by spectrophotometry at 260 nm and 280 nm absorbance. The ratio between the absorbances 260/280 nm was used to estimate the protein contamination. RNAs with a 260/280 nm ratio between 1.8 and 2.0 were considered to be good quality. Soon after, the complementary DNA was synthesized using the M-MLV reverse transcriptase, which promotes a complementary DNA strand from single-stranded RNA. The final part includes the real-time polymerase chain reaction (PCR), using the SYBR Green dye system (Sigma-Aldrich corporation, Saint Louis, MO, USA), which has a highly specific binding to double-stranded DNA, to detect the PCR product, as it accumulates during the reaction cycles.

#### 2.5.9. Tissue Morphology

To present representative images of the morphology of the articular tissues of each group (*n* = 3 animals/group), macroscopic images were captured using the stereomicroscope (Zeiss STEMI DV4, Göttingen, Germany) with a magnification of up to 32×. The image that best represents the profile of each group was selected by 2 separate researchers. In case of a disagreement, a third researcher was called in. The images of the profile of each group show the structures of both the femoral condyle and the tibial plateau. The images are purely representative.

#### 2.5.10. Histology

In the histological evaluations, the tissue samples from the right posterior joint (n-5 animals/group) were embedded in 10% paraformaldehyde (PFA) solution in 0.1 M phosphate buffer (pH 7.4) for 48 h. They were then embedded in paraffin after decalcification, dehydration, and clearing and sectioned into 5-μm-thick sections. The number of chondrocytes, the thickness of the cartilage, and the surface area of the cartilage were evaluated using hematoxylin–eosin (H&E) staining [71]. For each sample, 5 sections (5 slides) were produced and evaluated, and 5 images were taken from each section so that the results well represent the entire structure. The slides were read under a light microscope (up to 400× magnification) and quantified using the ImageJ program (version 1.54f). The number of chondrocytes was counted using the Cell Counter Plugin of the software, considering the nuclear staining of cells. To assess the degree of cartilage damage, the samples were stained with Alcian Blue and analyzed under a light microscope (100× magnification) according to the OARSI score described by Pritzker et al. [69]: 0—normal cartilage; 1—surface fibrillation, 2—fibrillation to deeper layers and chondrocyte disorientation, 3—fissures, 4—erosion of extracellular matrix components, 5—bone denudation, and 6—microfractures and bone remodeling. Histological analyses were performed by two blinded researchers. When results varied by more than 5% in the quantitative analyses, a new quantification was performed by a third researcher. In the OARSI score, any conflicting results were confirmed by the third researcher.

#### 2.5.11. Statistical Analysis

The data were presented as mean and standard deviation of the mean (SDM). The Shapiro–Wilk normality test was used to assess the normality of the data, and the one-way analysis of variance (ANOVA) was used to compare the groups. Tukey’s post hoc test was used for multiple comparisons, with a significance level of *p* < 0.05. The GraphPad Prism 7 software was used for statistical analysis.

## 3. Results

### 3.1. Metabolic Activity

Figure 4 shows the evaluation of activity levels of Krebs cycle and ETC components, including succinate dehydrogenase (SDH), citrate synthase (CS), complex I, and complex II. In the evaluation of SDH in A, there were no significant results. In B, both groups treated with moderate exercise showed a significant increase in CS activity compared to the OA group (*p*-value OA + 3× = 0.0189; *p*-value OA + 5× = 0.0487), with a high effect size (R^2^ = 0.59). Subsequently in C, Complex I showed a significant increase in activity in the OA + 3× group compared to the OA group (*p* = 0.05); however, there was an even greater significant increase in Complex I activity in the OA + 5× group compared to the OA group (*p* = 0.0002), with an effect size that is also high (R^2^ = 0.74). In D, in the analysis of Complex II, there were no significant results.

### 3.2. Weight Gain

In this analysis, the animals were weighed weekly. This procedure allowed us to determine the weight gain achieved by each animal at the end of the 8 weeks of training. In Figure 5, the OA + 5× group shows a lower weight gain compared to the sham group (*p* = 0.0409) and the OA group (*p* = 0.001) with a large effect size (R^2^ = 0.26).

### 3.3. Oxidative Stress

#### 3.3.1. Oxidants

To analyze aspects of oxidative stress, the activity of oxidant molecules was evaluated according to Figure 6. In A, the OA group showed a significant increase in DCF compared to the Sham group (*p* = 0.0056). Furthermore, in contrast to the OA group, there was a significant decrease in DCF in both the OA + 3× group (*p* = 0.0072) and especially in the OA + 5× group, with an even greater significant decrease (*p* = 0.0004) with a high effect size (R^2^ = 0.60). In B, the OA + 5× group showed no significant difference in nitrite levels compared to the sham group data. However, both the OA group (*p* = 0.0077) and the OA + 3× group (*p* = 0.0163) showed a significant increase in nitrite compared to the sham group and a high effect size (R^2^ = 0.44).

#### 3.3.2. Oxidative Damage and Antioxidants

The remaining aspects of the oxidative stress analysis evaluated oxidative damage (Figure 7A,B) and the activity of antioxidant enzymes (Figure 7C,D). In the carbonyl analysis in A, the OA + 5× group did not show a significant difference compared to the sham group, while both the OA (*p* = 0.0272) and OA + 3× groups (*p* = 0.0095) showed a significant increase in carbonyl compared to the sham group and presenting a high effect size (R^2^ = 0.46). In B, in the evaluation of sulfhydryl content, there were no significant results. During the evaluation of antioxidant activity in C, both exercise-treated groups showed a significant increase in SOD activity compared to the OA group (*p*-value OA + 3× = 0.0143; *p*-value OA + 5× = 0.0013), also presenting a high effect size (R^2^ = 0.41). In D, during the GSH analysis, there were no significant results.

### 3.4. Inflammatory Mediators

#### 3.4.1. Pro-Inflammatory Cytokines

An assessment of pro-inflammatory conditions was also performed using cytokines such as TNF-α, IL1-β, and IL6 (Figure 8). In A, the OA group showed a significant increase in TNF-α levels compared to the sham group (*p* < 0.0086). In contrast to the OA group, there was a significant decrease in TNF-α expression both in the OA + 3× group (*p* = 0.0032) and especially in the OA + 5× group with an even greater significant decrease in these markers (*p* = 0.0001), in addition to demonstrating a high effect size (R^2^ = 0.56). In B, the OA group showed a significant increase in IL1-β compared to the Sham group (*p* = 0.0002). In contrast, the OA + 3× (*p* = 0.0138) and OA + 5× (*p* < 0.0001) groups showed a significant decrease with a high effect size (R^2^ = 0.65) of this same cytokine compared to the OA group. In C, the OA group showed a significant increase in IL6 compared to the sham group (*p* = 0.0297). On the other hand, the exercise-treated groups showed no significant differences compared to the sham group, also showing a high effect size (R^2^ = 0.29).

#### 3.4.2. Anti-Inflammatory Cytokines

This study also evaluated the level of anti-inflammatory cytokines such as IL4, IL10, and the growth factor TGF-β, as shown in Figure 9. In A, the analysis of IL4 showed no significant results. In B, the OA + 5× group showed a significant increase in IL10 levels compared to the OA group (*p* = 0.0027) with a high effect size (R^2^ = 0.42). In C, the OA group showed significantly decreased expression of TGF-β compared to the sham group (*p* = 0.0134) and demonstrated a high effect size (R^2^ = 0.37).

### 3.5. Gene Expression

Figure 10 demonstrates the results of the quantification of BMP2 and MMP13 gene expression by PCR. In A, the OA group showed a significant decrease in BMP2 expression compared to the Sham group (*p* = 0.0391). On the other hand, the OA + 3× (*p* = 0.001) and OA + 5× (*p* = 0.0037) groups showed a significant increase with a high effect size (R^2^ = 0.68) compared to the AO group. In B, the AO group had a significant increase in MMP13 gene expression compared to the Sham group (*p* = 0.0343). However, the OA + 5× group showed a significant decrease compared to the AO group (*p* = 0.018) with a high effect size (R^2^ = 0.63).

### 3.6. Tissue Morphology

Figure 11 presents representative images of the articular morphological aspect related to the groups investigated in this study. The femoral condyles are shown in the upper part of the illustration, while the tibial plateaus of each group are shown in the lower part.

### 3.7. Histology

The analysis presented in Figure 12 shows evaluations demonstrated by representative histological images, lesion degrees using the OARSI score, chondrocyte quantification, cartilage thickness measurement, and contact surface measurement. In A, representative images of joints of each group are illustrated. In graph B, the OA group showed a significant increase in the degree of injury compared to the sham group (*p* = 0.0261). In contrast, both exercise treatment groups showed a significant decrease in injury degrees compared to the OA group (*p*-value OA + 3× = 0.025; *p*-value OA + 5× = 0.0473) with a high effect size (R^2^ = 0.52). In C, there were no significant results in the quantification of the number of chondrocytes. In D, the OA group showed a significant decrease in cartilage thickness compared to the sham group (*p* = 0.0267) with a large effect size (R^2^ = 0.69). On the other hand, both moderate exercise treatment groups showed no significant differences compared to the sham group. In E, the OA + 5× group showed a significant increase in contact surface area compared to the OA group (*p* = 0.0113) with a high effect size (R^2^ = 0.46).

## 4. Discussion

The majority of the protocols of the literature that aimed to treat OA through moderate exercise carried out at speeds between 12 and 18 m/min, for 30 min, for a period of 8 weeks, and frequencies between 3 and 5 times/week [4,10,11,12,13,14,30,31,32,33,34,35,36,37,38,39,40,41,42,43,44,45,46,47,48,49,50,51,52,53,54,55,56,57,58]. According to a review by Mazor et al. [48], it is also noteworthy that intervention periods with a speed of 18 m/min have positive morphological changes when implemented between 3 and 4 weeks. In this study, a pre-established exercise protocol with characteristics very similar to these findings in the literature was adapted, with the main difference being the weekly frequency of sessions in order to increase information on this characteristic of exercise in the treatment of osteoarthritis in animal models.

The results showed that exercise increased energy consumption in animals, especially in the group that performed 5 weekly exercise sessions, as analyzed by markers of complex I and citrate synthase (CS). CS is an important enzyme involved in the Krebs cycle, and increased activity indicates an elevated rate of mitochondrial energy production. Therefore, it has been used as a metabolic marker to assess respiratory and oxidative capacity [72,73]. The results of this study are aligned with the literature, especially due to the convergence between the indications of higher energy expenditure in the treated groups, which reaffirms the potential for metabolic intervention of the exercise protocols.

This is corroborated by the physiological data of lower weight gain in animals subjected to 5 weekly exercise sessions. The greater use of energy substrates prevents a positive energy balance, which may have contributed to the reduction in weight gain [74,75,76]. Since joint overload is one of the predisposing factors for the development of OA [2,8,9,76], it is estimated that the lower weight of these animals may have contributed to the decrease in mechanical stress on the knee joint and helped to prevent the progression of the disease in the OA + 5× group.

Further, in the analysis of markers of oxidative stress, both frequencies of moderate exercise used for OA treatment showed modulation of the evaluated parameters. However, the OA + 5× group, due to the higher number of sessions/weekly stimuli, may have contributed with greater efficiency to the modulation of the redox state.

Each moderate exercise session activates Nrf2 in response to increased ROS [74,77], i.e., the chronification of this activity, can generate environments with higher antioxidant activity [18,54,78]. As was identified in the results of the treated groups of this research, where an increase in antioxidant activity (SOD) was indicated.

On the other hand, the absence of treatment in the OA group demonstrated the effectiveness of the disease induction model, as it was able to promote the parameters expected for OA as evidenced by the evaluated markers. According to studies, the increase in reactive species is an aspect frequently associated with degeneration processes and consequently with OA [1,9,79]. It is noteworthy that the DCF marker has been used as an indicator of H_2_O_2_ [80]. Thus, the elevation of DCF in the OA group may be attributed to the increase in H_2_O_2_, reinforcing the osteoarthritic characteristics of these animals. On the other hand, the decrease in the treated groups highlights the chondroprotective role of exercise, mainly in the OA + 5× group, which obtained greater significance.

Evidence of the implication of reactive species in cartilage degradation also highlights the presence of nitrite as a marker for NO. in biological fluids of patients with OA and in cartilage in animal models of OA [81]. In addition, among the possible oxidative damages, this study highlights the susceptibility of protein oxidation in the presence of ROS and reactive nitrogen species (RNS), which could be verified based on the levels of elevated protein carbonyls and oxidation of protein thiols [80]. Thus, the significant increase in nitrite and carbonyl in the OA group indicates elevation in NO formation and protein oxidation, respectively, both attributes that point to the osteoarthritic profile [76]. Meanwhile, the absence of a significant increase in nitrite and carbonyl only in the OA + 5× group evidences the best antioxidant role of this exercise frequency [18,21,81].

The physiology of OA is also marked by a strong inflammatory role [82,83,84]. Among the inflammatory mediators in the pathogenesis of OA, studies highlight IL1-b, TNF-α, and IL6 as the most important regulators of pro-inflammatory processes [3,84,85]. Thus, the decrease or non-increase in these markers in the groups treated with different exercise frequencies reaffirms the anti-inflammatory role of exercise in both treatments, especially in the exercise therapy 5× per week which obtained more significant values.

This characteristic can be corroborated by the analysis of anti-inflammatory cytokines, which, mainly in the OA + 5× group, obtained a significant increase in IL10. This scenario indicates that the greater number of weekly exercise sessions induces a more persistent systemic anti-inflammatory response since, even performing euthanasia 72 h after the last exercise session, it was still possible to find increased levels of anti-inflammatory markers.

According to the literature, acute exercise (single session of aerobic or resistance exercise) promotes humoral changes from the secretion of cytokines, and its effects can last up to 72 h after the activity [14,17]. However, regular/chronic exercise (multiple sessions of acute exercise) is associated with cytokine-mediated changes even in the basal state [86]. This fact may have contributed to the modulations of pro-inflammatory markers in the OA + 3× group and of pro- and anti-inflammatory markers in the OA + 5× group.

These cytokines, also called exerkines, are signaling molecules released in response to exercise, which exert anti-inflammatory effects through endocrine, paracrine, and/or autocrine pathways. Among the known exerkines, IL6, IL10, and IL1-RA cytokines are particularly noteworthy [17]. It is also noteworthy that, although originally known for its pro-inflammatory role, IL6 is also recognized for its pleiotropic profile [16,17,18,74,86,87,88]. Having macrophages as secretors, IL6 induces a pro-inflammatory response associated with the pathogenesis of OA [3,71,84,85,88]. On the other hand, IL6 expressed by myocytes lasts a few hours after the end of exercise and leads to an anti-inflammatory response associated with the stimulation of the secretion of IL10 and IL1-RA [16,88,89].

Thus, as the analyses of this study were performed 72 h after the last exercise session, it is noteworthy that the IL6 evaluated here has a pro-inflammatory profile. Despite this, based on the literature, it is understood that each exercise session in this study promoted the release of IL6 in contracting myocytes, stimulating a systemic anti-inflammatory profile [6,16,17,18,74,87,88,89,90]. In this sense, contributing to the findings that evidenced a greater anti-inflammatory profile of the OA + 5× group due to a higher frequency of sessions.

To corroborate this perspective, the findings of the present study show the decrease in IL1-β in both treated groups, mainly in the OA + 5× group, an event that may be linked to the effect of exercise on the stimulation of IL1-RA. This cytokine binds to IL1-β receptors, inhibiting the pro-inflammatory action by not stimulating downstream cascades, in addition to blocking the binding of IL1-β to its receptor [16,74,90].

Additionally, another factor that may contribute to this perspective is the modulation of IL10 levels found in the groups with exercise, mainly in the OA + 5× group, which obtained a significant increase in this cytokine. IL10 has as its main function the negative regulation of adaptive immune responses and the minimization of tissue damage induced by pro-inflammatory effects [16,19,20]. In addition, IL10 is also related to the decrease in the expression of pro-inflammatory cytokines such as TNF-α [16,19], a circumstance that may have contributed to the more significant decrease in TNF-α in the OA + 5× group.

Corroborating the findings, a similar result was found in studies that evaluated treadmill exercise in OA in Wistar rats, in which a decrease in the pro-inflammatory markers IL1-β and TNF-α and an increase in the anti-inflammatory marker IL-10 were obtained [6,11]. Added to this, studies that used exercises to treat OA with other types of animals also found a decrease in IL1-β and TNF-α [60,62] and an increase in IL10 [37].

Furthermore, the literature indicates that exercise therapy can inhibit the production of MMP in chondrocytes and macrophages through the synthesis of IL10 [6,11,84]. Corroborating this scenario, an inversely proportional behavior of the IL10 data with the gene expression of MMP13 was evidenced, presenting inhibition of catabolic states in both groups, mainly in the OA + 5× group that obtained a significant decrease in MMP13. A similar aspect was visualized in a study with Wistar rats that analyzed the effects of moderate exercise on OA and the role of synoviocytes. In this research, an increase in IL10 and a consequent decrease in MMP13 were indicated [11]. In addition, other studies with different types of animals and OA models also reported an effect of decreasing MMP13 motivated by exercise [23,39,41,44].

Furthermore, the increase in the gene expression of BMP2 associated with roles of chondrogenesis, collagen, and proteoglycan production [91,92] evidences the stimulus to the anabolic role provided by exercise in both frequencies, in addition to indicating an attempt at tissue repair in the treated groups [12,23,92]. A similar result was found in a study that investigated different levels of exercise intensity on knee OA in Wistar rats. Moderate exercise suppressed cartilage degeneration while increasing BMP2, BMP4, and BMP6 [12]. Added to this, it was reported in another article that moderate exercise positively regulates the secretion of BMP2 and BMP6 by chondrocytes from the superficial region of cartilage and bone cells, conciliating with the prevention of osteoarthritic changes [23].

In addition, during the evaluation of TGF-β levels, a behavior of data similar to the modulations found in the IL10 data was also presented. It is estimated that the detection of highly expressed IL10 in the exercised groups signals the presence of M2 macrophages [6,84,92], which are involved in the release of several anti-inflammatory cytokines and associated growth factors, including TGF-β [76,85]. In this context, it is related that TGF-β may also suffer an increase in expression through higher frequencies of exercise and contribute to tissue repair and anti-inflammatory processes.

TGF-β promotes the production of ECM components in chondrocytes aiding in tissue repair. In particular, it is a potent inducer of lubricin in chondrocytes [91,93]. In turn, this evidence indicates that lubricin, also known as proteoglycan 4 (PRG4), is associated with a decrease in inflammatory processes mediated by TLR’s [82,94]. In addition to its lubricating function, this evidence indicates that lubricin can bind to pro-inflammatory receptors TLR2 and TLR4, inhibiting the activation of these receptors in a dose-dependent manner [82,94]. Furthermore, studies indicate that moderate exercise induces an increase in lubricin levels [11,95]. Thus, it is associated that the increase in TGF-β, through the stimulation of lubricin, can result in the inhibition of TLR’s and subsequent attenuation of their inflammatory responses in the joint, a scenario that could help to exemplify more significant findings in the decrease in pro-inflammatory markers in the OA + 5x group.

To investigate whether changes in tissue aspects were achieved, macro- and histological analyses were performed. In macroanalysis, the appearances of the cartilages denounce a protective effect of exercise, while it is possible to observe accentuated cartilage degradation in the OA group, while the groups treated with exercise imprint a structural aspect similar to the group without injury. These findings indicate an approximation with the results of biochemical and histological analysis, with confirmation also by the evaluation of the OARSI injury grades. In this histological scale, the groups with exercise presented to prevent the progression of OA from the low levels of injury grades; data also reconciled with the non-decrease in cartilage thickness in both groups with therapy [71]. Additionally, the OA + 5× group also presented tissue adaptation according to the increase in the surface contact measurement. In this way, it is assumed that in this group the pressure exerted on the cartilage is reduced by a larger contact area, favoring a greater distribution of mechanical forces on the cartilage tissue.

Regarding the number of chondrocytes, due to cartilage damage recorded in the histological data of injury grades and cartilage thickness of the OA group, there was possibly cell proliferation in such a way that it approached the levels between the groups. This is a perspective that goes against the literature when it states that one of the measures to combat biomechanical and/or biochemical damage to cartilage results in an increase in the number of cells in an attempt to repair the injury, but that due to tissue limitations, it can end up generating senescence with subsequent apoptosis of chondrocytes [3,76,85]. Judging by the persistent inflammatory and oxidative stress picture evidenced in the findings of the present study, it is estimated that if the experiment were conducted for a longer time, it would possibly lead to cell death in the aforementioned group.

Another aspect of moderate physical exercise involves the mechanical action of movement that, with the exception of adequate proportions, stimulates a greater diffusion of synovial fluid in the cartilage, providing more nutrients and oxygen for chondrocytes [11,23]. With this, although both therapies show modulation of biochemical and histological parameters, it is estimated that higher weekly exercise frequencies can provide a more persistent nutritive environment for chondrocytes, stimulating better local homeostasis. This is corroborated through the general conjuncture of the parameters investigated, with the best results evidenced in the OA + 5× group (Figure 13).

Based on the assumption that physical exercise can exert anti-inflammatory, antioxidant, and anabolic effects systemically [6,16,17,74,87,88,89,90], we hypothesized that increasing the weekly exercise frequency potentiated these beneficial effects in the experimental model of osteoarthritis in Wistar rats. Thus, the exercise frequency performed five times a week promoted greater modulation of oxidative stress and the inflammatory response, in addition to better preservation of articular cartilage, when compared to the protocol with a lower frequency (3×/week).

## 5. Conclusions

Considering the aspects listed in this study, both treatments with different frequencies of moderate treadmill exercise showed attenuation in the development processes of OA, signaling the protective effect of cartilage from the use of exercise as therapy. However, from the analyzed parameters, the frequency of 5× weekly sessions of moderate exercise is evidenced as the dosage with the greatest number of beneficial results for the treatment of OA. The greater number of exercise sessions proved to modulate the redox state, stimulate the reduction in pro-inflammatory and catabolic markers, and induce anti-inflammatory and anabolic states, in addition to inhibiting the tissue progression of OA disease.

Finally, this study has some limitations that should be considered when interpreting the results. First, the experimental design was performed exclusively in an animal model of chemically induced osteoarthritis with all-male rats, which, although widely validated, does not fully reproduce the complexity of the disease in humans. Furthermore, the analyses were conducted at a single time point after the intervention, preventing us from observing potential late effects or the behavior of variables during the course of treatment. Assessment of functional and behavioral outcomes, such as pain analysis through sensitivity testing and spontaneous activity monitoring, should be integrated into future studies to correlate molecular findings with clinical improvement. The animals in this study maintained a double-support gait, dividing the load on the joint between the affected lower limb and the contralateral upper limb. Therefore, extrapolation to single-leg gait in humans requires confirmation through clinical studies.

## Figures and Tables

**Figure 1 biology-14-01537-f001:**
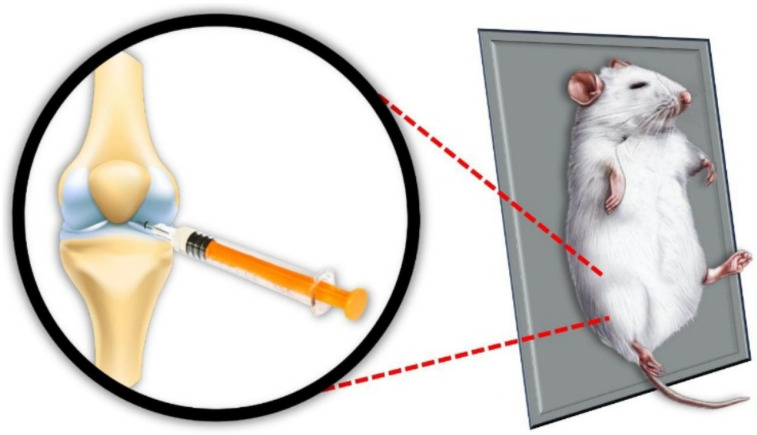
Illustrative image of the OA induction model by intra-articular injection of MIA. Source: Own authorship.

**Figure 2 biology-14-01537-f002:**
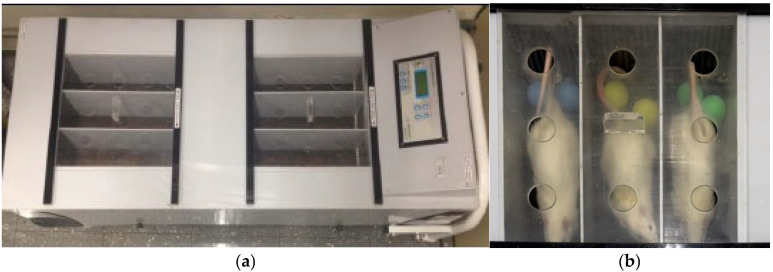
Instrument for performing exercise protocols. (**a**), Treadmill used for exercise by Wistar rats. (**b**), Exercise performance on a treadmill using tennis balls to stimulate animals. Source: Own authorship.

**Figure 3 biology-14-01537-f003:**
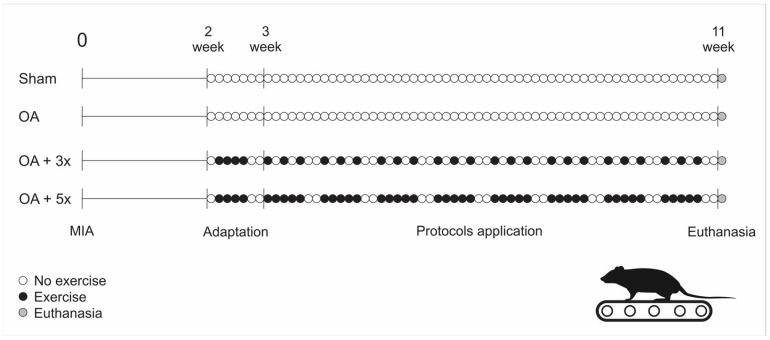
Treadmill exercise frequency protocol. Source: Own authorship.

**Figure 4 biology-14-01537-f004:**
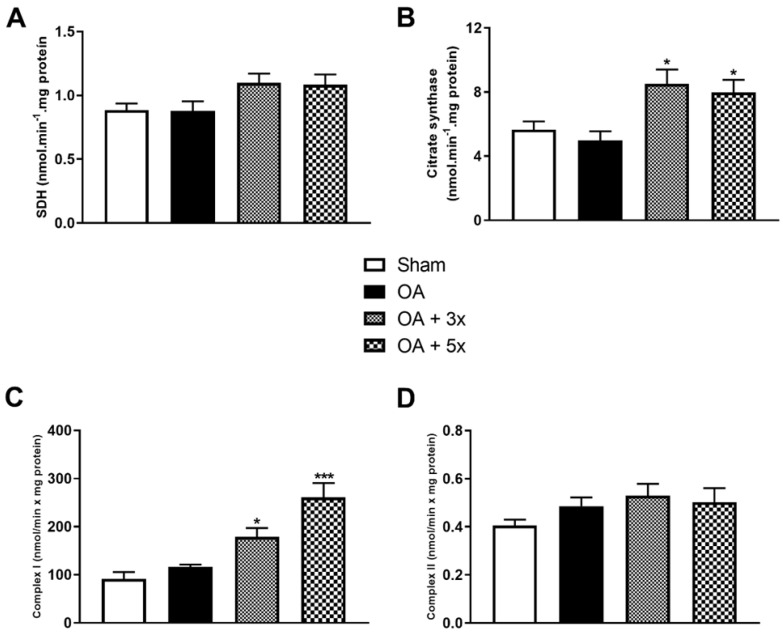
Effects of moderate exercise treatment performed 3 or 5 times per week on metabolic activity. The graphs show evaluations of SDH in (**A**), CS in (**B**), Complex I in (**C**), and Complex II in (**D**). These evaluations were performed on samples from the gastrocnemius muscle. Abbreviations: SDH, succinate dehydrogenase; CS, citrate synthase. Data are presented as mean ± SDM where * *p* < 0.05 vs. OA group; *** *p* < 0.001 vs. OA group; (One-way ANOVA followed by Tukey’s post hoc test).

**Figure 5 biology-14-01537-f005:**
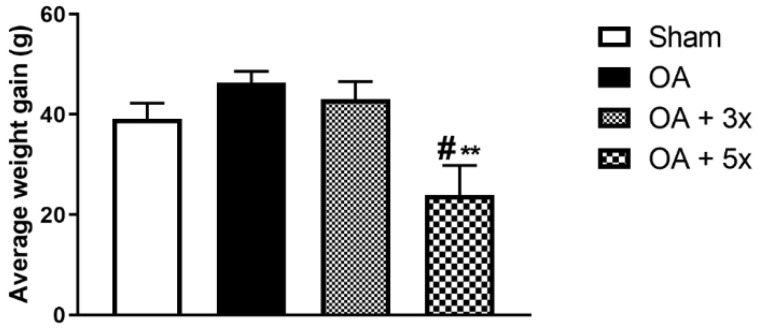
Effects of moderate exercise treatment performed 3 or 5 times per week on average body weight gain. This evaluation was performed based on the total body mass gain of the animals. Abbreviations: g, grams. Data are presented as mean ± SDM where # *p* < 0.05 vs. sham group; ** *p* < 0.01 vs. OA group; (One-way ANOVA followed by Tukey’s post hoc test).

**Figure 6 biology-14-01537-f006:**
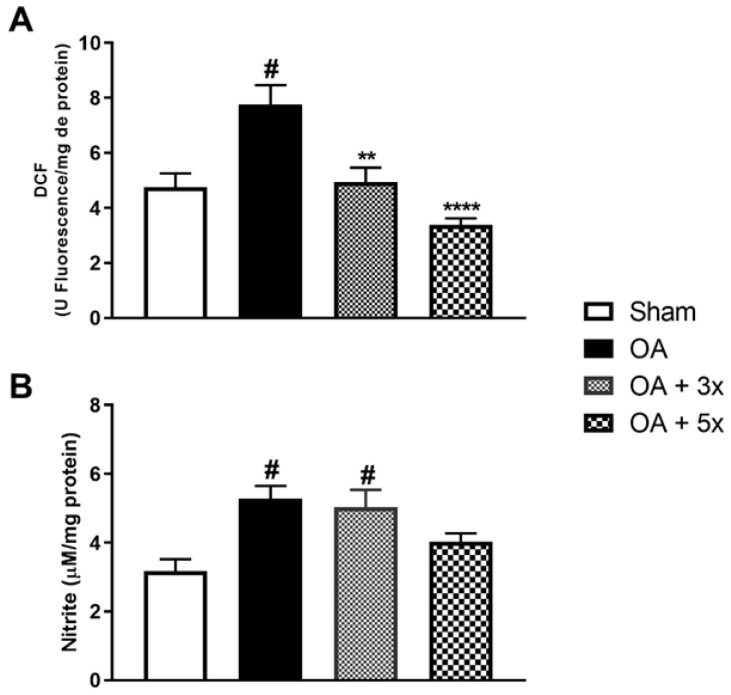
Effects of moderate exercise treatment performed 3 or 5 times a week on oxidant levels. The graphs show the evaluations of DCF in (**A**) and Nitrite in (**B**). These evaluations were performed from samples of the intracapsular tissues of the knee joint. Abbreviations: DCF, Dichlorofluorescein. Data are presented as mean ± SDM where # *p* < 0.05 vs. sham group; ** *p* < 0.01 vs. OA group; **** *p* < 0.0001 vs. OA group; (One-way ANOVA followed by Tukey’s post hoc test).

**Figure 7 biology-14-01537-f007:**
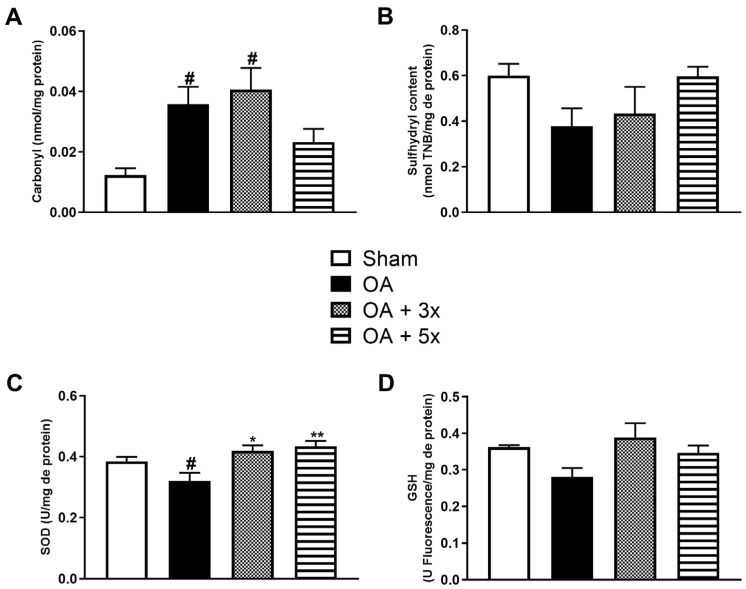
Effects of moderate exercise treatment performed 3 or 5 times a week on oxidative damage and antioxidant levels. The graphs show the evaluations of Carbonyl in (**A**), Sulfhydryl content in (**B**), SOD in (**C**), and GSH in (**D**). These evaluations were performed from samples of the intracapsular tissues of the knee joint. Abbreviations: SOD, superoxide dismutase; GSH, glutathione. Data are presented as mean ± SDM where # *p* < 0.05 vs. sham group; * *p* < 0.05 vs. OA group; ** *p* < 0.01 vs. OA group; (One-way ANOVA followed by Tukey’s post hoc test).

**Figure 8 biology-14-01537-f008:**
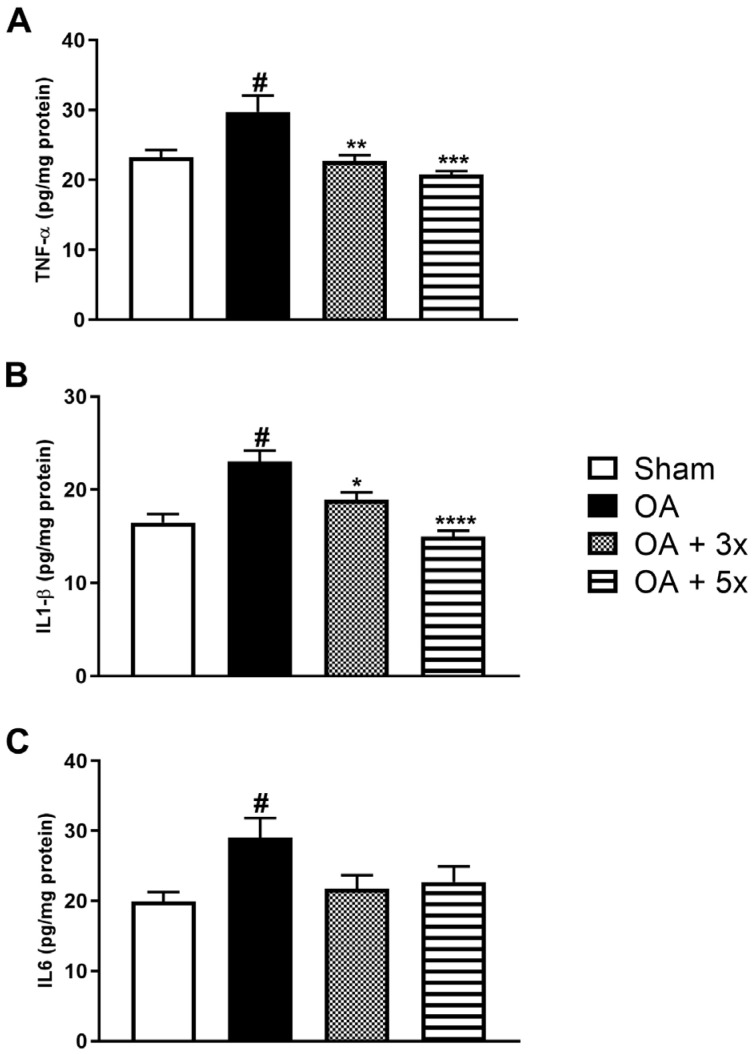
Effects of moderate exercise treatment performed 3 or 5 times a week on pro-inflammatory cytokine levels. The graphs show the evaluations of TNF-α in (**A**), IL1-β in (**B**), and IL6 in (**C**). These evaluations were performed using samples from the intracapsular tissues of the knee joint. Abbreviations: IL, interleukin; TNF, tumour necrosis factor. Data are presented as mean ± SDM where # *p* < 0.05 vs. sham group; * *p* < 0.05 vs. OA group; ** *p* < 0.01 vs. OA group; *** *p* < 0.001 vs. OA group; **** *p* < 0.0001 vs. OA group; (One-way ANOVA followed by Tukey’s post hoc test).

**Figure 9 biology-14-01537-f009:**
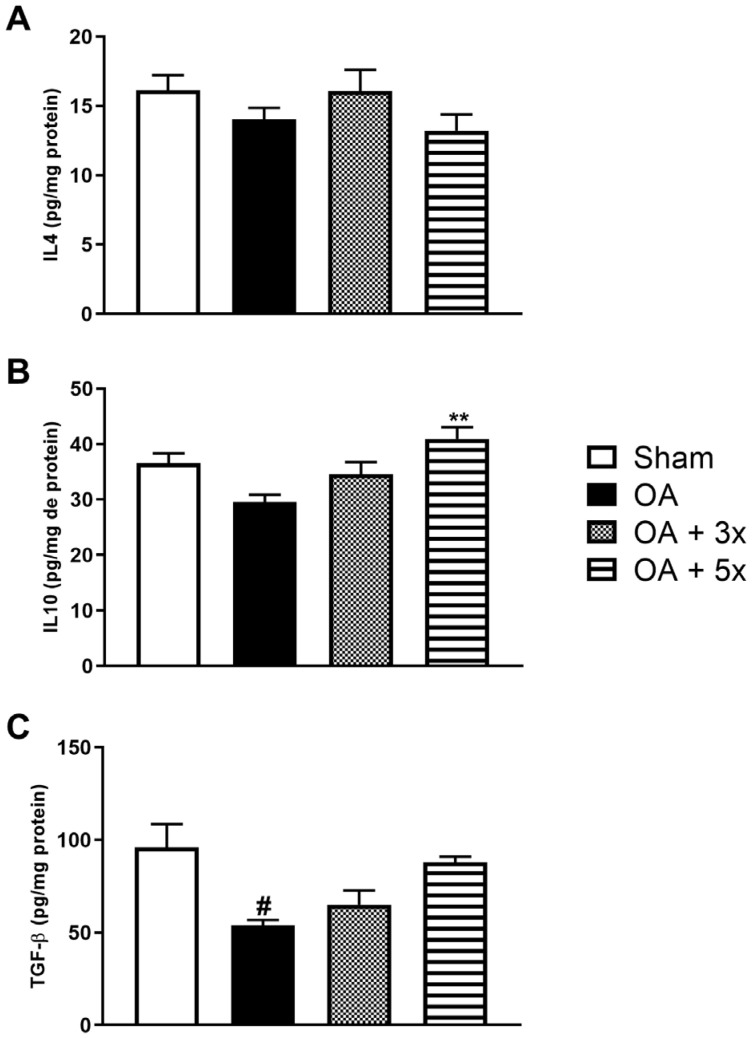
Effects of moderate exercise treatment performed 3 or 5 times a week on anti-inflammatory cytokine levels. The graphs show the evaluations of IL4 in (**A**), IL10 in (**B**), and TGF-β in (**C**). These evaluations were performed using samples from the intracapsular tissues of the knee joint. Abbreviations: IL, interleukin; TGF, transforming growth factor. Data are presented as mean ± SDM where # *p* < 0.05 vs. sham group; ** *p* < 0.01 vs. OA group; (One-way ANOVA followed by Tukey’s post hoc test).

**Figure 10 biology-14-01537-f010:**
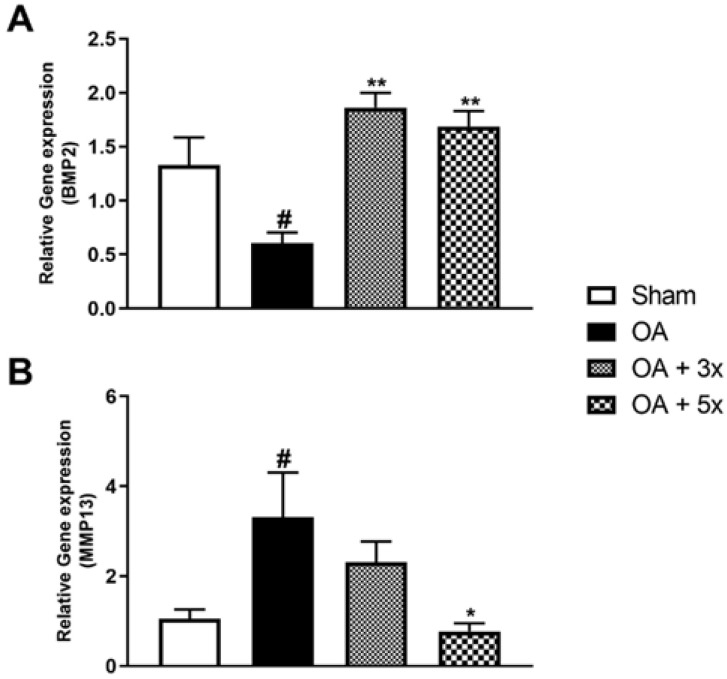
Effects of moderate exercise treatment performed 3 or 5 times a week on BMP2 and MMP13 gene expression. The graphs show the evaluations of BMP2 in (**A**) and MMP13 in (**B**). These evaluations were performed using samples from the intracapsular tissues of the knee joint. Abbreviations: BMP, bone morphogenetic protein; MMP, matrix metalloproteinase. Data are presented as mean ± SDM where # *p* < 0.05 vs. sham group; * *p* < 0.05 vs. OA group; ** *p* < 0.01 vs. OA group; (One-way ANOVA followed by Tukey’s post hoc test).

**Figure 11 biology-14-01537-f011:**
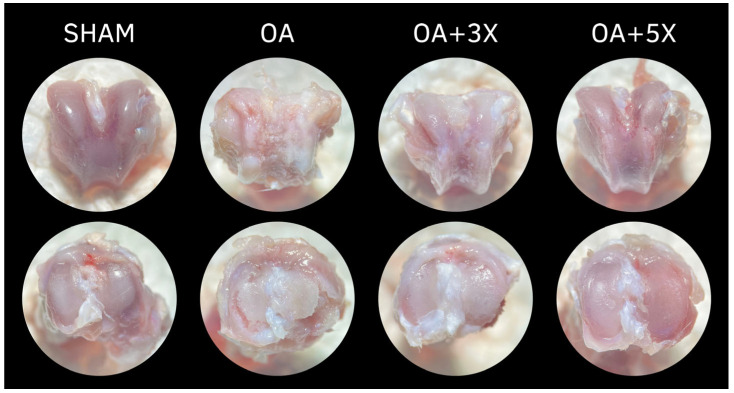
Effects of moderate exercise treatment performed 3 or 5 times a week on the morphological aspect of the articular tissues of the femoral condyle (**upper part**) and tibial plateau (**lower part**).

**Figure 12 biology-14-01537-f012:**
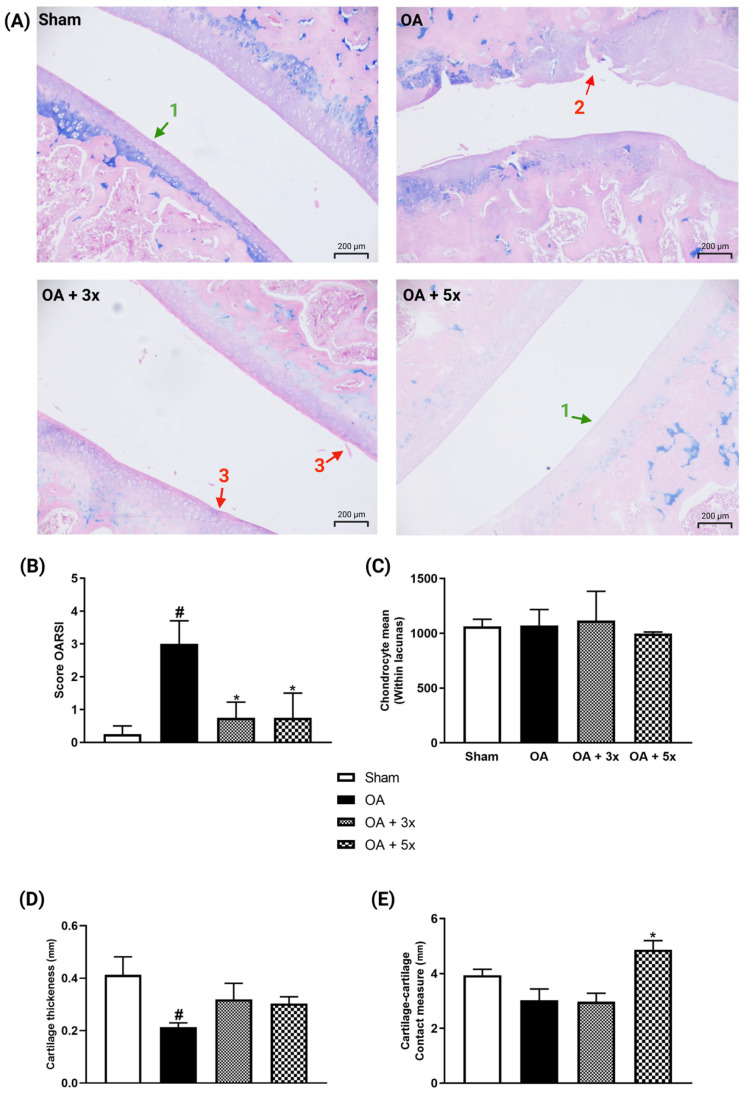
Effects of moderate exercise treatment performed 3 or 5 times a week on histological analysis. (**A**) Representative images of each group stained with Alcian Blue (100× magnification). The arrows marked with number 1 point to the intact surface. Arrow number 2 shows a complex vertical fissure associated with an erosive process extending to the middle third. Arrow number 3 shows a discrete superficial fibrillation. (**B**) OARSI score. (**C**) Number of chondrocytes. (**D**) Cartilage thickness. (**E**) Contact surface area measurement. These evaluations were performed using samples from the articular tissues of the tibial plateau and femoral condyle. Abbreviations: OARSI, Osteoarthritis Research Society International. Data are presented as mean ± SDM where # *p* < 0.05 vs. sham group; * *p* < 0.05 vs. OA group; (One-way ANOVA followed by Tukey’s post hoc test).

**Figure 13 biology-14-01537-f013:**
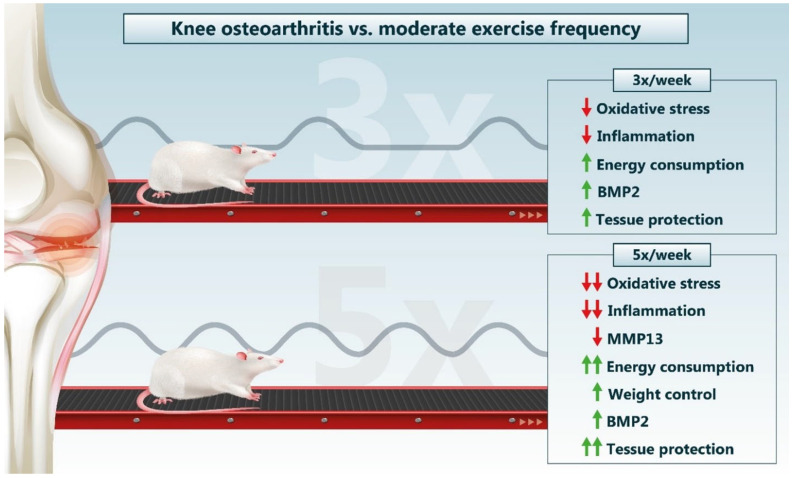
Summary of the main findings on different frequencies of moderate exercise as a treatment for knee OA. Red arrows indicate decreases and green arrows indicate increases. Source: Own authorship.

**Table 1 biology-14-01537-t001:** Summary table of the moderate exercise protocol on a treadmill.

Week	Frequency (Weekly)	Time (min)	Speed (m/min)
Adaptation	4×	10 to 25	10
1	3× or 5×	30	13
2	3× or 5×	30	13
3	3× or 5×	30	13
4	3× or 5×	30	13
5	3× or 5×	30	16
6	3× or 5×	30	16
7	3× or 5×	30	16
8	3× or 5×	30	16

Source: Adapted from Cifuentes et al. [54].

## Data Availability

The data that support the findings of this study are available from the corresponding author, [Silveira, PC], upon reasonable request.

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
