# Peer review of "Exercise as Osteoarthritis Treatment in Wistar Rats Promotes Frequency-Dependent Benefits"

_biology, 2025, doi:10.3390/biology14111537_

Round 1

Reviewer 1 Report

Comments and Suggestions for Authors

Overall impression: This manuscript asks a worthwhile, clinically relevant question — whether the frequency of an exercise regimen produces different structural, biochemical and pain-related outcomes in an MIA-induced osteoarthritis model in Wistar rats — and the authors combine behavioral, histologic and biochemical endpoints in a single controlled study. The topic and approach are appropriate and potentially useful to the field, but the paper in its current form needs substantive revision before it can be recommended for publication.

Strengths: The multi-endpoint design (behavior + histology + biochemistry) and the direct comparison of multiple exercise frequencies are clear assets. If the results are robust and reproducible, they could provide practical guidance for preclinical exercise-dosing studies and help bridge to clinical recommendations.

Major concerns and required clarifications: The Methods lack important details needed to judge experimental rigor and reproducibility. Please state how animals were randomized to groups and whether experimenters were blinded during behavioral testing, tissue processing and histologic scoring. Provide an a priori sample-size/power justification or, at minimum, a rationale for the chosen group sizes and the final N analyzed per endpoint (after any exclusions). Detail the MIA protocol (concentration, volume, injection technique, timing), animal characteristics (sex, age, weight range), housing conditions and analgesia. For the exercise protocol, explain how intensity was defined and calibrated, how compliance was monitored, and whether sessions were supervised. For biochemical assays, add catalog numbers, detection limits and replicate strategy; for histology, report stain protocols, magnification/objective, number of sections and fields analyzed per joint, the scoring system used, and inter-rater reliability metrics if scoring was subjective.

Statistics and interpretation: The manuscript should report normality testing, exact p-values, effect sizes with 95% confidence intervals, the statistical tests used for each comparison and any correction for multiple testing, and the software/version used. At present the conclusions are broadly consistent with the data direction, but given the gaps in methodological and quantitative reporting the strength of the claims is uncertain. I recommend the authors temper definitive language until methodological transparency and full quantitative histology are provided (phrase conclusions as “frequency-dependent trends” unless robust statistical evidence supports stronger claims).

Figures, tables and histology (urgent): Figure legends must state group N, the statistical test, the definition of error bars and exact p-values. Plot individual animal datapoints overlaid on summary statistics to show variability. Provide raw biochemical and histology data as supplemental files. Figure 12 requires specific corrections: add clear panel labels (A–D) on each micrograph, include a readable scale bar with µm units (or a single calibrated scale bar on the final panel with an explicit legend statement if magnification is identical), report objective and final magnification, specify stain and processing details, declare any uniform image adjustments, annotate key features with arrows and define them in the legend, and present accompanying quantitative histology scores (blinded scoring, n per group, mean ± SD and individual datapoints).

References and framing: References are generally suitable but the Introduction and Discussion should better situate the work within recent animal and clinical studies on exercise dosing in osteoarthritis; temper claims of novelty to reflect an incremental but useful contribution unless the authors can point to a clear and unique mechanistic insight.

Author Response

REVIEWER 1

Overall impression: This manuscript asks a worthwhile, clinically relevant question — whether the frequency of an exercise regimen produces different structural, biochemical and pain-related outcomes in an MIA-induced osteoarthritis model in Wistar rats — and the authors combine behavioral, histologic and biochemical endpoints in a single controlled study. The topic and approach are appropriate and potentially useful to the field, but the paper in its current form needs substantive revision before it can be recommended for publication.

Strengths: The multi-endpoint design (behavior + histology + biochemistry) and the direct comparison of multiple exercise frequencies are clear assets. If the results are robust and reproducible, they could provide practical guidance for preclinical exercise-dosing studies and help bridge to clinical recommendations.

Major concerns and required clarifications: The Methods lack important details needed to judge experimental rigor and reproducibility. Please state how animals were randomized to groups and whether experimenters were blinded during behavioral testing, tissue processing and histologic scoring. Provide an a priori sample-size/power justification or, at minimum, a rationale for the chosen group sizes and the final N analyzed per endpoint (after any exclusions). Detail the MIA protocol (concentration, volume, injection technique, timing), animal characteristics (sex, age, weight range), housing conditions and analgesia. For the exercise protocol, explain how intensity was defined and calibrated, how compliance was monitored, and whether sessions were supervised. For biochemical assays, add catalog numbers, detection limits and replicate strategy; for histology, report stain protocols, magnification/objective, number of sections and fields analyzed per joint, the scoring system used, and inter-rater reliability metrics if scoring was subjective.

R: Dear reviewer, thank you for your thorough review of our article. We have followed your suggestions for modifications to improve the article.

The animals were all numbered and randomly assigned to the experimental group. The following excerpt has been added to the text:

“The rats were numbered and randomly divided, by drawing lots, into 4 groups containing 14 to 16 animals”

Furthermore, the researchers were blinded for most of the experiment. Osteoarthritis induction and animal weighing were performed by blinded researchers. During the training period, there was no blinding, as the investigators needed to know the groups to which the rats belonged to place them on the treadmill. The removal of the structures for analysis and the analyses themselves (biochemical, molecular, and histological) were also performed by blinded researchers. The following excerpts have been added to the text to clarify the blinding:

 “The applicator was blinded throughout the model induction.”

“Subsequently, with the researchers blinded, the intra-articular tissues and the gastrocnemius muscle of the right hindlimb were removed, numbered and stored in a -80°C freezer for subsequent analysis.”

“To ensure reproducibility and accuracy of results, all analyses were performed blindly and in duplicate. The values ​​presented correspond to the average of the replicates.”

An explanation regarding the sample n was added to the text, in the methodology section:

“Based on the instructions issued by animal ethics committees, animal research recommends the use of the 3Rs: Replacement, Reduction, and Refinement [1]. Thus, to use the minimum number of animals necessary to obtain the results, the number of animals was based on a review of studies with animal models and exercise [2-40] for the possibility of a difference of up to 17 to 25% in the parameters to be analyzed between the groups, with a variance of up to 10%, calculated with the EDA tool [41-42]. This results in a sample size of 14 animals per group (6 to 9 animals for metabolic activity analyses, ELISAs, and biochemical tests and 5 animals/group for histological analyses).

It was also implemented to make it clearer about the exclusion of animals:

“Not all animals adapt to walking on the treadmill, resulting in their exclusion from the experiment.”

“The excluded animals were not used in the analyses.”

As requested, more details about the MIA protocol, animal characteristics, housing conditions, and analgesia were added to the methodology section. The following sections were added:

“The procedure was performed with a single intra-articular injection of 1.5 mg of sodium monoiodoacetate (MIA) (Sigma-Aldrich corporation, lot: #SLCK0393) diluted in 50 μl of 0.9% saline solution [29], at a concentration of 30 mg/mL.”

“The animals were anesthetized with 4% isoflurane via inhalation until the corneal-palpebral reflex was absent and movement was absent. After being anesthetized, they were placed in the supine position, their knees were flexed and trichotomized, aseptically cleaned, and the MIA compound was administered through the medial region adjacent to the infrapatellar ligament of the right knee.”

It should be noted that for greater methodological transparency of the procedure, the region where the procedure is performed is illustrated in figure 1.

“Syringes with 4 mm fixed needles and a capacity of 500μl, commonly used for insulin administration, were used. To better control the amount injected, 50 μL of the preparation was aspirated from the Falcon tube into the syringes, and the syringes were changed after each injection.”

Additional Notes

Time: Single application of the compound prepared immediately before model induction.

Animal Characteristics: These characteristics are incorporated in item "2.1 Animals." However, for greater clarity regarding the animals' housing, we have added more information about their storage locations.

“These boxes (cages) were approximately 30 cm long x 20 cm wide x 13 cm high. They had a wood shaving lining the interior base and a metal grate on top for air circulation, inspection, and storage of food and water bottles.”

Analgesia: for model induction and treatment protocol, the animals were not subjected to analgesia, as the use of analgesic measures could interfere with the results by reducing pain, one of the cardinal signs of the inflammatory process and osteoarthritis itself, compromising the reproducibility of the model.

Regarding the exercise protocol:

The exercise intensity proposed in this study was based on the pre-established protocol by Cifuentes et al. [43], which recommends a moderate intensity for Wistar rats and is reaffirmed in the literature by the study by Leandro et al. [44], which evaluated VO2 consumption in treadmill exercise protocols with Wistar rats. Furthermore, the literature review conducted for the study [2-40] corroborates this by recommending a speed of 12 to 18 m/min as a moderate intensity for treadmill exercise in studies involving animal models of osteoarthritis and exercise as a form of treatment.

The animals' participation was stimulated by placing table tennis balls behind the animals, which, when touched or made sounds, kept them active.To improve the transparency of information, we also added the following to the text:

“It should be noted that to eliminate any eventualities that could lead to the interruption of the exercise, regardless of the cause, the animals were monitored by the researchers during all sessions.”

“The equipment had electronic speed, time, and electrostimulation regulators, all controlled by a digital panel that displayed speed (m/min), time (minutes), distance (meters), and amperage (mA). The tilt was manually adjusted and remained steady throughout the experiment. Before the protocols began, the equipment underwent inspection and calibration.”

We also made changes to the written methodology for biochemical assays. Detection limits and more information about the replication strategy were added. The following sections were added to the methodology:

“To ensure reproducibility and accuracy of results, all analyses were performed blindly and in duplicate. The values ​​presented correspond to the average of the replicates.”

“(detection limit = 0.01 U fluorescence/mg protein).”

“(detection limit = 0.1 uM/mg protein)”

“(detection limit = 0.001 nmol TNB/mg protein)”

“(detection limit = 0.1 U/mg protein)”

“(detection limit = 0.025 U fluorescence/mg protein)”

“(detection limit: IL1-β, IL6 and TGF-β = 8 pg/mg protein; TNF-α = 15.6 pg/mg protein; IL10 = 32 pg/mg protein; IL4 = 4 pg/mg protein).”

“(detection limit = 0.005 nmol/mg protein)”

            In histology, further details of the methodology were clarified, as suggested. The following excerpts were added:

“When results varied by more than 5% in the quantitative analyses, a new quantification was performed by a third researcher. In the OARSI score, any conflicting results were confirmed by the third researcher.”

“For each sample, 5 sections (5 slides) were produced and evaluated, and 5 images were taken from each section so that the results well represent the entire structure.”

Statistics and interpretation: The manuscript should report normality testing, exact p-values, effect sizes with 95% confidence intervals, the statistical tests used for each comparison and any correction for multiple testing, and the software/version used. At present the conclusions are broadly consistent with the data direction, but given the gaps in methodological and quantitative reporting the strength of the claims is uncertain. I recommend the authors temper definitive language until methodological transparency and full quantitative histology are provided (phrase conclusions as “frequency-dependent trends” unless robust statistical evidence supports stronger claims).

R: Dear reviewer, thank you for your constructive comment. As suggested, we have included the exact p-values ​​and effect sizes in the results, and specified the statistical tests used for each comparison. Furthermore, we have added information in the methodology section that all analyses were performed using GraphPad Prism 7 software.

Figures, tables and histology (urgent): Figure legends must state group N, the statistical test, the definition of error bars and exact p-values. Plot individual animal datapoints overlaid on summary statistics to show variability. Provide raw biochemical and histology data as supplemental files. Figure 12 requires specific corrections: add clear panel labels (A–D) on each micrograph, include a readable scale bar with µm units (or a single calibrated scale bar on the final panel with an explicit legend statement if magnification is identical), report objective and final magnification, specify stain and processing details, declare any uniform image adjustments, annotate key features with arrows and define them in the legend, and present accompanying quantitative histology scores (blinded scoring, n per group, mean ± SD and individual datapoints).

R: Dear reviewer, the exact p-values ​​and effect size have been added to the results text. We believe that adding them to the figure legends would make them too long and would repeat information already provided in the text. As suggested, the statistical test used has been added to all legends.

Regarding the individual data points in the graphs: we believe the graphs are clearer and easier to interpret for the reader as they are. Furthermore, the bar above the graphs demonstrates the standard deviation of the results for each group, allowing for an interpretation of the data variability.

In Figure 12, the corrections were made as suggested. Histological analyses were performed with n=5 animals/group. Exact p-values ​​were added to the results text. A legible scale with µm units was added to all images. Arrows pointing to features assessed on the OARSI scale were also added to the images, and the captions were supplemented with more information. The caption is as follows:

“Figure 12. Effects of moderate exercise treatment performed 3 or 5 times a week on histological analysis. The figure show in A: Representative images of each group stained with Alcian Blue (100x magnification). The arrows marked with number 1 point to the intact surface. Arrow number 2 shows a complex vertical fissure associated with an erosive process extending to the middle third. Arrow number 3 shows a discrete superficial fibrillation; in B: OARSI score; in C: Number of chondrocytes; in D: Cartilage thickness; in E: contact surface area measurement. These evaluations were performed using samples from the articular tissues of the tibial plateau and femoral condyle. Abbreviations: OARSI, Osteoarthritis Research Society International. Data are presented as mean +/- SDM where: #p<0.05 vs. sham group; *p<0.05 vs. OA group; **p<0.01 vs. OA group; (One-way ANOVA followed by Tukey's post hoc test).”

It's important to note that the representative images are magnified at 100x to provide a more comprehensive view of the joint's condition. However, chondrocyte counts were performed at 400x magnification, as described in the methodology.

References and framing: References are generally suitable but the Introduction and Discussion should better situate the work within recent animal and clinical studies on exercise dosing in osteoarthritis; temper claims of novelty to reflect an incremental but useful contribution unless the authors can point to a clear and unique mechanistic insight.

R: Dear reviewer, changes were made to the introduction and discussion as indicated. Additional points were raised regarding studies addressing exercise protocols, confirming the lack of information in the literature on exercise frequency in animal models. Claims of novelty were moderated, attempts were made to summarize the discussion, and additional studies that could contribute to the results were highlighted. At the same time, limitations of the study and recommendations for future studies were added to the end of the discussion.

Added to the introduction:

“In addition, within the discussion of the beneficial effects of exercise, the mainte-nance/chronification of physical exercise is listed as an important point for obtaining favorable effects [18-23]. That said, the frequency of exercise execution can be deter-minant to potentiate protective effects in the treatment of OA [10]. Thus, in mid-2021, a search of the PubMED-MEDLINE database was conducted using MESH descriptors such as "Osteoarthritis AND Exercise" and filters such as "Other animals" and "10 years," yielding 81 articles. After reading the titles and abstracts, 43 articles were se-lected. After reading the full text, 38 studies met the eligibility criteria (articles that associated animal models of knee OA with physical exercise over the last decade). It was found that none of the 38 studies found analyzed the effect of different weekly ex-ercise frequencies on OA.” [2-4, 6-40].

“The same lack of information on frequency was found in review studies on the same topic that evaluated articles from before the last decade. In these reviews, the main outcome was based on exercise intensity.” [45-48]

Very repetitive topics have been removed from the discussion to improve readability, and the following has been added to the discussion:

“The majority of the protocols of the literature that aimed to treat OA through moderate exercise carried out at speeds between 12 and 18 m/min, for 30 min, for a period of 8 weeks, and frequencies between 3 and 5 times/week [2-40]. According to a review by Mazor et al. [47], it is also noteworthy that interven-tion periods with a speed of 18 m/min have positive morphological changes when im-plemented between 3 and 4 weeks. However, for longer intervention periods, lower speeds such as 13 to 15 m/min are recommended. In this study, a pre-established exercise protocol with characteristics very similar to these findings in the literature was adapted, with the main difference being the weekly frequency of sessions in order to increase information on this characteristic of exercise in the treatment of osteoarthritis in animal models.”

“Corroborating the findings, a similar result was found in studies that evaluated treadmill exercise in OA in Wistar rats, in which a decrease in the pro-inflammatory markers IL1-β and TNF-α [4-5] and an increase in the anti-inflammatory marker IL-10 were obtained [4-5]. Added to this, studies that used exercises to treat OA with other types of animals also found a decrease in IL1-β and TNF-α [12] and an increase in IL10 [16].”

“Finally, this study has some limitations that should be considered when interpreting the results. First, the experimental design was performed exclusively in an animal model of chemically induced osteoarthritis with all-male rats, which, although widely validated, does not fully reproduce the complexity of the disease in humans. Furthermore, the analyses were conducted at a single time point after the intervention, preventing us from observing potential late effects or the behavior of variables during the course of treatment. Assessment of functional and behavioral outcomes, such as pain analysis through sensitivity testing and spontaneous activity monitoring, should be integrated into future studies to correlate molecular findings with clinical improvement. The animals in this study maintained a double-support gait, dividing the load on the joint between the affected lower limb and the contralateral upper limb. Therefore, extrapolation to single-leg gait in humans requires confirmation through clinical studies.”

Reviewer 2 Report

Comments and Suggestions for Authors

I thank you for the opportunity to review this article. A really very interesting topic, in addition to being a well-written text. Bellow just some points that should be considerate.

Add study’s hypotheses.

Does any sample calculation was conducted?

At section Animals is not clear how many animals each group have. At the first Paragraph is mentioned “divided into groups of 4 to 5” and at the second paragraph the authors reported “divided into 4 groups containing 14 to 16 animals”. Please clarify this point.

According with authors “However, for longer intervention periods, lower speeds such as 13 to 15 m/min are recommended”. Why the 16 speed was used?

Do animals compare before protocol?

Add study limitations and suggestion for future studies.

Author Response

REVIEWER 2

I thank you for the opportunity to review this article. A really very interesting topic, in addition to being a well-written text. Bellow just some points that should be considerate.

Add study’s hypotheses.

The following hypothesis was added:

“Based on the assumption that physical exercise can exert anti-inflammatory, antioxidant, and anabolic effects systemically [5, 49-55], we hypothesize that increasing the weekly frequency of exercise potentiates these beneficial effects in the experimental model of osteoarthritis in Wistar rats. Thus, we hypothesize that the frequency of exercise performed five times per week promotes greater modulation of oxidative stress and the inflammatory response, in addition to better preservation of articular cartilage, when compared to the protocol with a lower frequency (3x/week).”

Does any sample calculation was conducted?

R: An explanation regarding the sample n was added to the text, in the methodology section:

“Based on the instructions issued by animal ethics committees, animal research recommends the use of the 3Rs: Replacement, Reduction, and Refinement [1]. Thus, to use the minimum number of animals necessary to obtain the results, the number of animals was based on a review of studies with animal models and exercise [2-40] for the possibility of a difference of up to 17 to 25% in the parameters to be analyzed between the groups, with a variance of up to 10%, calculated with the EDA tool [41-42]. This results in a sample size of 14 animals per group (6 to 9 animals for metabolic activity analyses, ELISAs, and biochemical tests and 5 animals/group for histological analyses).

At section Animals is not clear how many animals each group have. At the first Paragraph is mentioned “divided into groups of 4 to 5” and at the second paragraph the authors reported “divided into 4 groups containing 14 to 16 animals”. Please clarify this point.

R: Indeed, the original text could have caused confusion regarding the distribution of the animals. The passage was revised to clarify that sixty Wistar rats were initially used, allocated to four experimental groups, each containing 14 to 16 animals. The previous mention of "groups of 4 to 5" referred only to the number of animals kept per cage, not the size of the experimental groups. This information has now been rewritten in the manuscript to ensure greater clarity and accuracy.

According with authors “However, for longer intervention periods, lower speeds such as 13 to 15 m/min are recommended”. Why the 16 speed was used?

R:       According to the review by Mazor et al. [47], speeds of 18 m/min or higher promote cartilage degradation and the progression of OA. Therefore, for the present study, the pre-established protocol by Cifunetes et al. [43] was used, which used a speed of 16 m/min. Furthermore, studies that used exercise as a treatment for OA indicated a moderate intensity variability of 12 to 18 m/min for treadmill exercise.

Do animals compare before protocol?

R: In this study, the animals were not compared individually before and after treatment. The experimental design adopted was a comparative type between independent groups, in which different sets of animals were subjected to different experimental conditions (Sham, OA, OA + 3x, and OA + 5x).

Therefore, the analyses were performed by comparing the results obtained between the groups after the experimental period, allowing us to evaluate the differential effects of the exercise protocols on the osteoarthritis model at the same time point. This design is widely used in studies with animal models.

Furthermore, although the animals were not previously compared, the sample was selected based on species, weight, sex, and age to ensure a more homogeneous sample.

Add study limitations and suggestion for future studies.

R: Dear reviewer, as suggested, we have added the main limitations of the study and some suggestions for future research at the end of the discussion. The following excerpt has been added:

“This study has some limitations that should be considered when interpreting the results. First, the experimental design was performed solely in an animal model of chemically induced osteoarthritis, which, although widely validated, does not fully reproduce the complexity of the disease in humans. Furthermore, the analyses were conducted in a single time interval after the intervention, preventing us from observing possible late effects or the behavior of variables during the course of treatment. Assessment of functional and behavioral outcomes, such as pain analysis through sensitivity testing and monitoring of spontaneous activity, should be integrated into future studies to correlate molecular findings with clinical improvement. The animals in this study maintained a double-support gait, dividing the load on the joint between the affected lower limb and the contralateral upper limb. Therefore, ex-trapolation to single-leg gait in humans requires confirmation through clinical studies.”

Reviewer 3 Report

Comments and Suggestions for Authors

I would like to congratulate the authors on an excellent experiment and scientific text. The manuscript is very well written and interesting. I believe that only minor changes could be made, without compromising the content of the text:

lines 50 and 51, remove the initials of the authors cited.
line 100, line 111, present the volumetric units.
line 126, clearly indicate the author of reference 30.
line 129-135 - I understand that this section is discussion, not methods.
line 300 - it would be better to present in standard deviation and not in standard error, since this is sample data and not population data.
line 621 - stick to only answering the objectives in the conclusion.

Author Response

REVIEWER 3

I would like to congratulate the authors on an excellent experiment and scientific text. The manuscript is very well written and interesting. I believe that only minor changes could be made, without compromising the content of the text:

lines 50 and 51, remove the initials of the authors cited.

line 100, line 111, present the volumetric units.

line 126, clearly indicate the author of reference 30.

line 129-135 - I understand that this section is discussion, not methods.

line 300 - it would be better to present in standard deviation and not in standard error, since this is sample data and not population data.

line 621 - stick to only answering the objectives in the conclusion.

Dear reviewer, we sincerely thank you for your positive review of our work and for your constructive suggestions. All requested modifications have been made to the revised version of the manuscript. The following changes were made:

  • The initials of the authors cited in lines 50 and 51 have been removed, leaving only their last names.
  • Volumetric units have been added to lines 100 and 111 to ensure clarity and consistency.
  • The author of reference 30 has been clearly indicated in line 126 to improve citation accuracy.
  • The text previously present in lines 129-135 has been moved to the beginning of the discussion.
  • "Standard error of the mean" has been changed to "standard deviation of the mean" throughout the text, as suggested. The standard deviation was used for statistics. Only one minor typo has already been corrected.

Reviewer 4 Report

Comments and Suggestions for Authors

The manuscript entitled “Exercise as Osteoarthritis Treatment in Wistar Rats Promotes Frequency-Dependent Benefits” investigates the role of treadmill exercise frequency in modulating the progression of osteoarthritis in an established rat model. Overall, the topic is highly relevant, as the study addresses an important gap in the literature, namely the effect of exercise frequency in comparison with the already well-described influences of exercise intensity. The manuscript is generally well-structured, with a clear rationale, detailed methodology, and extensive biochemical, histological, and molecular analyses. The results are consistent and convincingly demonstrate that moderate exercise five times per week provides superior protective and therapeutic effects compared with three weekly sessions. The findings contribute to the growing body of evidence supporting physical activity as a cornerstone in osteoarthritis management.

Nevertheless, there are several aspects that could be improved to enhance clarity and impact. The introduction is comprehensive, but at times repetitive, and would benefit from a more concise presentation of the known effects of exercise on OA and a sharper focus on why frequency remains an understudied but critical variable. The methods are described in remarkable detail, which ensures reproducibility, but some sections (particularly biochemical assays) could be streamlined or moved to supplementary material. The figures are generally clear and well-labeled, although some panels could benefit from higher resolution and more explicit legends to facilitate interpretation. The statistical approach appears appropriate, yet the sample size justification could be discussed more explicitly, particularly regarding power calculations and the decision to increase group size by two animals initially.

The results are robust and demonstrate internal consistency across metabolic, inflammatory, oxidative stress, and histological domains. However, the discussion is very long, occasionally reiterating results instead of critically integrating them with the broader literature. A more concise and focused discussion emphasizing translational implications would improve readability. It would be useful for the authors to acknowledge more explicitly the limitations of the study, such as the use of only male rats, the short duration of the protocol, and the inherent challenges of extrapolating findings from animal models to human osteoarthritis. Furthermore, while the authors highlight the dose–response effect of exercise, it would be important to comment on the potential upper limits of exercise frequency or intensity that could lead to deleterious effects, to avoid overgeneralization.

The reference list is extensive and up to date, although there is some redundancy in citing multiple studies for similar concepts. A careful re-evaluation of references to ensure conciseness and relevance would improve the flow. The English language is generally clear but could benefit from minor editing for grammar and style, as some sentences are overly long or complex.

In conclusion, this is a scientifically sound and well-conducted study that provides novel insights into the frequency-dependent benefits of exercise in osteoarthritis management. With revisions to improve conciseness in the introduction and discussion, a clearer emphasis on translational relevance, and minor editing of language and figures, the manuscript has the potential to make a valuable contribution to the field. I recommend minor revisions before acceptance.

--------------------------

Extend Report:

Summary of the manuscript
The manuscript entitled “Exercise as Osteoarthritis Treatment in Wistar Rats Promotes Frequency-Dependent Benefits” investigates the therapeutic role of moderate treadmill exercise in a rat model of knee osteoarthritis induced by sodium monoiodoacetate. Sixty animals were allocated to four groups (Sham, OA, OA+3x, OA+5x), and the study compared the effects of three versus five weekly exercise sessions. The authors evaluated metabolic activity, oxidative stress, inflammatory mediators, histological changes, and gene expression. The results suggest that both exercise protocols mitigated OA progression, with more pronounced benefits in the higher-frequency group, including reduced oxidative and inflammatory markers, improved cartilage morphology, and enhanced anabolic signaling.
Evaluation of methodology, analyses, and conclusions The methodology is generally appropriate, using a well-established MIA-induced OA model and including a reasonable number of animals per
group. The exercise protocol was consistent with prior literature, and the range of molecular, biochemical, and histological assessments provides a comprehensive view of outcomes. Nevertheless, several aspects would benefit from greater clarification. The description of randomization and blinding procedures could be more explicit to confirm the minimization of bias. Some of the biochemical assays are reported with limited methodological details, which may hinder reproducibility. The statistical analyses (ANOVA with Tukey post-hoc) are appropriate, though a more detailed justification for sample size calculation would strengthen the rigor. The results are clearly presented, supported by figures and tables, but some legends are overly concise and could provide more interpretative context. The conclusions are consistent with the findings, though they could better acknowledge the limitations of extrapolating animal model data to human osteoarthritis.

Constructive feedback for the authors
1.
Expand the methodology to include clearer information on randomization, blinding, and sample size calculation.
2.
Provide additional details in the figure legends to guide interpretation of the results.
3.
In the discussion, more explicitly acknowledge the translational limitations of animal studies and emphasize the need for validation in human clinical trials.
4.
Strengthen the contextualization of findings by comparing them to other recent preclinical and clinical studies on exercise in OA.
5.
Consider reducing redundancy in the discussion, which occasionally reiterates points without adding new insights.

In conclusion, this is an interesting and well-conceived study that
provides new data on the frequency-dependent benefits of exercise in OA management. With some clarifications and improvements in presentation, the manuscript would represent a valuable contribution to the field.

Author Response

REVIEWER 4

The manuscript entitled “Exercise as Osteoarthritis Treatment in Wistar Rats Promotes Frequency-Dependent Benefits” investigates the role of treadmill exercise frequency in modulating the progression of osteoarthritis in an established rat model. Overall, the topic is highly relevant, as the study addresses an important gap in the literature, namely the effect of exercise frequency in comparison with the already well-described influences of exercise intensity. The manuscript is generally well-structured, with a clear rationale, detailed methodology, and extensive biochemical, histological, and molecular analyses. The results are consistent and convincingly demonstrate that moderate exercise five times per week provides superior protective and therapeutic effects compared with three weekly sessions. The findings contribute to the growing body of evidence supporting physical activity as a cornerstone in osteoarthritis management.

Nevertheless, there are several aspects that could be improved to enhance clarity and impact. The introduction is comprehensive, but at times repetitive, and would benefit from a more concise presentation of the known effects of exercise on OA and a sharper focus on why frequency remains an understudied but critical variable. The methods are described in remarkable detail, which ensures reproducibility, but some sections (particularly biochemical assays) could be streamlined or moved to supplementary material. The figures are generally clear and well-labeled, although some panels could benefit from higher resolution and more explicit legends to facilitate interpretation. The statistical approach appears appropriate, yet the sample size justification could be discussed more explicitly, particularly regarding power calculations and the decision to increase group size by two animals initially.

R: Dear reviewer, as per your suggestions, the introduction has been changed to make it more concise and clear.

We appreciate your feedback on our methodology, but we chose to maintain the description of the experimental details. We believe this is essential for reproducibility, and other reviewers requested that even more details be added.

Regarding the figures: The figures and figures have been changed to provide more information and make them clearer to readers.

A better explanation of the sample size calculation was also added to the methodology section:

 “Based on the instructions issued by animal ethics committees, animal research recommends the use of the 3Rs: Replacement, Reduction, and Refinement [1]. Thus, to use the minimum number of animals necessary to obtain the results, the number of animals was based on a review of studies with animal models and exercise [2-40] for the possibility of a difference of up to 17 to 25% in the parameters to be analyzed between the groups, with a variance of up to 10%, calculated with the EDA tool [41-42]. This results in a sample size of 14 animals per group (6 to 9 animals for metabolic activity analyses, ELISAs, and biochemical tests and 5 animals/group for histological analyses).

The results are robust and demonstrate internal consistency across metabolic, inflammatory, oxidative stress, and histological domains. However, the discussion is very long, occasionally reiterating results instead of critically integrating them with the broader literature. A more concise and focused discussion emphasizing translational implications would improve readability. It would be useful for the authors to acknowledge more explicitly the limitations of the study, such as the use of only male rats, the short duration of the protocol, and the inherent challenges of extrapolating findings from animal models to human osteoarthritis. Furthermore, while the authors highlight the dose–response effect of exercise, it would be important to comment on the potential upper limits of exercise frequency or intensity that could lead to deleterious effects, to avoid overgeneralization.

R:       As suggested by the reviewer, the discussion was revised and summarized, removing some redundant paragraphs and sentences. However, it should be noted that after adding the initial section on studies addressing exercise protocols, new corroborations of the findings, and especially on the study's limitations, the discussion remained close to its original length. However, this also allowed us to adjust the conclusion to be shorter and more objective.

The following paragraph on the study's limitations was added:

“Finally, this study has some limitations that should be considered when interpret-ing the results. First, the experimental design was performed exclusively in an animal model of chemically induced osteoarthritis with all-male rats, which, although widely validated, does not fully reproduce the complexity of the disease in humans. Further-more, the analyses were conducted at a single time point after the intervention, pre-venting us from observing potential late effects or the behavior of variables during the course of treatment. Assessment of functional and behavioral outcomes, such as pain analysis through sensitivity testing and spontaneous activity monitoring, should be integrated into future studies to correlate molecular findings with clinical improve-ment. The animals in this study maintained a double-support gait, dividing the load on the joint between the affected lower limb and the contralateral upper limb. Therefore, extrapolation to single-leg gait in humans requires confirmation through clinical studies.”

The reference list is extensive and up to date, although there is some redundancy in citing multiple studies for similar concepts. A careful re-evaluation of references to ensure conciseness and relevance would improve the flow. The English language is generally clear but could benefit from minor editing for grammar and style, as some sentences are overly long or complex.

R:       Citations and references were extensively revised, with some being removed as suggested by the reviewer. However, to address new questions raised by the reviewers, some references needed to be added to the text.

Furthermore, the entire English version was proofread, and minor edits were made to improve the article's overall writing.

In conclusion, this is a scientifically sound and well-conducted study that provides novel insights into the frequency-dependent benefits of exercise in osteoarthritis management. With revisions to improve conciseness in the introduction and discussion, a clearer emphasis on translational relevance, and minor editing of language and figures, the manuscript has the potential to make a valuable contribution to the field. I recommend minor revisions before acceptance.

R: Dear reviewer, we appreciate all your contributions and suggestions aimed at improving our article. We find them all very relevant and strive to address them in the best way possible so that the article can be published.

--------------------------

Extend Report:

Summary of the manuscript

The manuscript entitled “Exercise as Osteoarthritis Treatment in Wistar Rats Promotes Frequency-Dependent Benefits” investigates the therapeutic role of moderate treadmill exercise in a rat model of knee osteoarthritis induced by sodium monoiodoacetate. Sixty animals were allocated to four groups (Sham, OA, OA+3x, OA+5x), and the study compared the effects of three versus five weekly exercise sessions. The authors evaluated metabolic activity, oxidative stress, inflammatory mediators, histological changes, and gene expression. The results suggest that both exercise protocols mitigated OA progression, with more pronounced benefits in the higher-frequency group, including reduced oxidative and inflammatory markers, improved cartilage morphology, and enhanced anabolic signaling.

Evaluation of methodology, analyses, and conclusions The methodology is generally appropriate, using a well-established MIA-induced OA model and including a reasonable number of animals per

group. The exercise protocol was consistent with prior literature, and the range of molecular, biochemical, and histological assessments provides a comprehensive view of outcomes. Nevertheless, several aspects would benefit from greater clarification. The description of randomization and blinding procedures could be more explicit to confirm the minimization of bias. Some of the biochemical assays are reported with limited methodological details, which may hinder reproducibility. The statistical analyses (ANOVA with Tukey post-hoc) are appropriate, though a more detailed justification for sample size calculation would strengthen the rigor. The results are clearly presented, supported by figures and tables, but some legends are overly concise and could provide more interpretative context. The conclusions are consistent with the findings, though they could better acknowledge the limitations of extrapolating animal model data to human osteoarthritis.

Constructive feedback for the authors

  1. Expand the methodology to include clearer information on randomization, blinding, and sample size calculation.

Completed as suggested. To improve transparency, additional details have been added, such as:

“The rats were numbered and randomly divided, by drawing lots, into 4 groups containing 14 to 16 animals”

“The applicator was blinded throughout the model induction.”

“Subsequently, with the researchers blinded, the intra-articular tissues and the gastrocnemius muscle of the right hindlimb were removed, numbered and stored in a -80°C freezer for subsequent analysis.”

To ensure reproducibility and accuracy of results, all analyses were performed blindly and in duplicate. The values ​​presented correspond to the average of the replicates.

An explanation regarding the sample n was added to the text, in the methodology section:

“Based on the instructions issued by animal ethics committees, animal research recommends the use of the 3Rs: Replacement, Reduction, and Refinement [1]. Thus, to use the minimum number of animals necessary to obtain the results, the number of animals was based on a review of studies with animal models and exercise [2-40] for the possibility of a difference of up to 17 to 25% in the parameters to be analyzed between the groups, with a variance of up to 10%, calculated with the EDA tool [41-42]. This results in a sample size of 14 animals per group (6 to 9 animals for metabolic activity analyses, ELISAs, and biochemical tests and 5 animals/group for histological analyses).

It was also implemented to make it clearer about the exclusion of animals:

“Not all animals adapt to walking on the treadmill, resulting in their exclusion from the experiment.”

“The excluded animals were not used in the analyses.”

  1. Provide additional details in the figure legends to guide interpretation of the results.

R: As suggested by the reviewers, all results legends have had information added to facilitate interpretation.

  1. In the discussion, more explicitly acknowledge the translational limitations of animal studies and emphasize the need for validation in human clinical trials.

R: Made as suggested by the reviewer. Placed in the last paragraph of the discussion:

“This study has some limitations that should be considered when interpreting the results. First, the experimental design was performed solely in an animal model of chemically induced osteoarthritis, which, although widely validated, does not fully reproduce the complexity of the disease in humans. Furthermore, the analyses were conducted in a single time interval after the intervention, preventing us from observing possible late effects or the behavior of variables during the course of treatment. Assessment of functional and behavioral outcomes, such as pain analysis through sensitivity testing and monitoring of spontaneous activity, should be integrated into future studies to correlate molecular findings with clinical improvement. The animals in this study maintained a double-support gait, dividing the load on the joint between the affected lower limb and the contralateral upper limb. Therefore, ex-trapolation to single-leg gait in humans requires confirmation through clinical studies.”

  1. Strengthen the contextualization of findings by comparing them to other recent preclinical and clinical studies on exercise in OA.

R:       Rewritten as suggested by the reviewer. Additional points were raised regarding studies addressing exercise protocols, confirming the lack of information in the literature on exercise frequency in animal models. Furthermore, additional studies that could contribute to the results were highlighted.Adicionado à introdução:

“In addition, within the discussion of the beneficial effects of exercise, the mainte-nance/chronification of physical exercise is listed as an important point for obtaining favorable effects [18-23]. That said, the frequency of exercise execution can be deter-minant to potentiate protective effects in the treatment of OA [10]. Thus, in mid-2021, a search of the PubMED-MEDLINE database was conducted using MESH descriptors such as "Osteoarthritis AND Exercise" and filters such as "Other animals" and "10 years," yielding 81 articles. After reading the titles and abstracts, 43 articles were se-lected. After reading the full text, 38 studies met the eligibility criteria (articles that associated animal models of knee OA with physical exercise over the last decade). It was found that none of the 38 studies found analyzed the effect of different weekly ex-ercise frequencies on OA.” [2-4, 6-40].

“The same lack of information on frequency was found in review studies on the same topic that evaluated articles from before the last decade. In these reviews, the main outcome was based on exercise intensity.” [45-48]

Added to discussion:

“The majority  of the protocols of the literature that aimed to treat OA through moderate exercise carried out at speeds between 12 and 18 m/min, for 30 min, for a period of 8 weeks, and frequencies between 3 and 5 times/week [2-40]. According to a review by Mazor et al. [47], it is also noteworthy that interven-tion periods with a speed of 18 m/min have positive morphological changes when im-plemented between 3 and 4 weeks. However, for longer intervention periods, lower speeds such as 13 to 15 m/min are recommended. In this study, a pre-established exercise protocol with characteristics very similar to these findings in the literature was adapted, with the main difference being the weekly frequency of sessions in order to increase information on this characteristic of exercise in the treatment of osteoarthritis in animal models.”

“Corroborating the findings, a similar result was found in studies that evaluated treadmill exercise in OA in Wistar rats, in which a decrease in the pro-inflammatory markers IL1-β and TNF-α [4-5] and an increase in the anti-inflammatory marker IL-10 were obtained [4-5]. Added to this, studies that used exercises to treat OA with other types of animals also found a decrease in IL1-β and TNF-α [12] and an increase in IL10 [16].”

  1. Consider reducing redundancy in the discussion, which occasionally reiterates points without adding new insights.

As suggested by the reviewer, the article was revised and summarized, removing some redundant paragraphs and sentences. However, it should be noted that after adding the initial section on studies addressing exercise protocols, new corroborations of the findings, and especially on the study's limitations, the discussion remained close to its original length. However, this also allowed us to adjust the conclusion to a shorter and more objective length.

In conclusion, this is an interesting and well-conceived study that

provides new data on the frequency-dependent benefits of exercise in OA management. With some clarifications and improvements in presentation, the manuscript would represent a valuable contribution to the field.